# *activin-2* is required for regeneration of polarity on the planarian anterior-posterior axis

**Jennifer K. Cloutier**[1,2,3,4], **Conor L. McMann**[1,2,3], **Isaac M. Oderberg**[1,2,3], **Peter W. Reddien**[1,2,3]*

**1** Whitehead Institute for Biomedical Research, Cambridge, MA, United States of America, **2** Department of Biology, Massachusetts Institute of Technology, Cambridge, MA, United States of America, **3** Howard Hughes Medical Institute, Chevy Chase, MD, United States of America, **4** Harvard/MIT MD-PhD, Harvard Medical School, Boston, MA, United States of America

* reddien@wi.mit.edu

**Data Availability Statement:** All data is deposited and available at NCBI GSE148376.

**Funding:** This work was supported by the Eleanor Schwartz Charitable Foundation (to PWR). PWR is an investigator of HHMI. The funders had no role in

## Abstract

Planarians are flatworms and can perform whole-body regeneration. This ability involves a mechanism to distinguish between anterior-facing wounds that require head regeneration and posterior-facing wounds that require tail regeneration. How this head-tail regeneration polarity decision is made is studied to identify principles underlying tissue-identity specification in regeneration. We report that inhibition of *activin-2*, which encodes an Activin-like signaling ligand, resulted in the regeneration of ectopic posterior-facing heads following amputation. During tissue turnover in uninjured planarians, positional information is constitutively expressed in muscle to maintain proper patterning. Positional information includes Wnts expressed in the posterior and Wnt antagonists expressed in the anterior. Upon amputation, several wound-induced genes promote re-establishment of positional information. The head-versus-tail regeneration decision involves preferential wound induction of the Wnt antagonist *notum* at anterior-facing over posterior-facing wounds. Asymmetric activation of *notum* represents the earliest known molecular distinction between head and tail regeneration, yet how it occurs is unknown. *activin-2* RNAi animals displayed symmetric wound-induced activation of *notum* at anterior- and posterior-facing wounds, providing a molecular explanation for their ectopic posterior-head phenotype. *activin-2* RNAi animals also displayed anterior-posterior (AP) axis splitting, with two heads appearing in anterior blastemas, and various combinations of heads and tails appearing in posterior blastemas. This was associated with ectopic nucleation of anterior poles, which are head-tip muscle cells that facilitate AP and medial-lateral (ML) pattern at posterior-facing wounds. These findings reveal a role for Activin signaling in determining the outcome of AP-axis-patterning events that are specific to regeneration.

study design, data collection and analysis, decision to publish, or preparation of the manuscript.

**Competing interests:** The authors have declared that no competing interests exist.

## Author summary

A central problem in animal regeneration is how animals determine what body part to regenerate. Planarians are flatworms that can regenerate any missing body region, and are studied to identify mechanisms underlying regeneration. At transverse amputation planes, a poorly understood mechanism specifies regeneration of either a head or a tail. This head-versus-tail regeneration decision-making process is referred to as regeneration polarity and has been studied for over a century to identify mechanisms that specify what to regenerate. The gene *notum*, which encodes a Wnt antagonist, is induced within hours after injury preferentially at anterior-facing wounds, where it specifies head regeneration. We report that Activin signaling is required for regeneration polarity, and the underlying asymmetric activation of *notum* at anterior- over posterior-facing wounds. We propose that Activin signaling is involved in regeneration-specific responses broadly in the animal kingdom.

## Introduction

The planarian *Schmidtea mediterranea* is a powerful model for the study of whole-body regeneration. In many planarian species a single animal can be cut into multiple small fragments, which can each regenerate a complete animal within a matter of weeks [1,2]. Planarian regeneration involves new cell production from a population of stem cells called neoblasts. Neoblasts produce all new cell types in planarian regeneration and also allow extensive tissue turnover in uninjured animals [3]. Planarian tissue patterning requires the regionalized expression of numerous signaling ligands and their pathway components [3]. Many such genes are termed position control genes (PCGs) and are defined by displaying constitutive regional expression and a patterning RNAi phenotype, or association with a planarian-patterning pathway. PCGs are predominantly expressed in planarian muscle [4].

The distribution of tissues on the planarian anterior-posterior (AP) axis and the head-versus-tail regeneration decision at transverse amputation planes prominently involve Wnt signaling [5–18]. A Wnt expression and activity gradient exists along the planarian AP axis, with the anterior being Wnt-signaling low and the posterior being Wnt-signaling high [5–18]. Multiple planarian *wnt* family genes are considered PCGs and are predominantly expressed in muscle [4]. Inhibition of the Wnt pathway, such as with *β-catenin-1* RNAi, causes regeneration of posterior-facing heads [5–7]. *β-catenin-1* RNAi during tissue turnover causes ectopic anterior PCG expression in the posterior and loss of posterior PCG expression, resulting in ectopic head formation [5–7,10,13,15,16,18].

Regeneration following transverse amputation of the (AP) axis requires a mechanism to specify formation of a head or a tail at the amputation plane–this determination is made within hours following amputation and involves wound signaling [9,10,12,19]. Wounds generically induce the transcription of ~200+ genes [19,20]. *wnt1* is activated at all wounds and promotes tail regeneration at posterior-facing wounds. The gene *notum* is unique among wound-induced genes in that it is preferentially induced at anterior-facing wounds [12,19]. *notum* encodes a broadly conserved Wnt deacylase that inhibits Wnt signaling [21], and is required for the regeneration of a head instead of a tail at anterior-facing wounds [12]. The activation of these two genes results in a Wnt-inhibited anterior-facing wound for head regeneration and a Wnt-active posterior-facing wound for tail regeneration. Wound-induced *notum* expression occurs specifically in preexisting longitudinal muscle fibers, which are oriented along the AP axis [22]. After this wound-induced phase of *notum* expression, other anterior PCGs are

expressed in muscle and progenitors for the anterior pole are specified. The anterior pole is a signaling center [23–26] comprised of muscle cells generated from neoblasts during regeneration [23–25], and the anterior pole also expresses *notum* [12]. Thus the two phases of *notum* expression in head regeneration involve distinct cells; i.e., the wound-induced *notum*⁺ cells do not directly form the anterior pole [12,23,25]. The asymmetric wound induction of *notum* is the earliest known difference in gene expression that exists between wounds that go on to make a head instead of a tail [12,19]. This step is therefore central to regeneration, yet, how the asymmetric activation of *notum* at anterior-versus-posterior-facing wounds is accomplished is unknown.

Additional wound-induced genes have roles in determining the outcome of planarian regeneration [9,27–29]. *follistatin* is activated at planarian wounds and is required for a set of cellular and molecular responses collectively referred to as the missing tissue response [27,28,30]. The missing tissue response involves cell proliferation at the wound, a body-wide increase in cell death, and sustained expression of wound-induced genes. *follistatin* is also required for regeneration of anterior PCG expression domains and head regeneration [27,28,30], but only when amputations are made in a Wnt-high environment (i.e., transverse amputation in the trunk and tail) [30]. *follistatin* RNAi leads to the upregulation of wound-induced *wnt1* at 6 hours post amputation (hpa)–higher wound-induced *wnt1* levels in an already high Wnt environment are proposed to block the ability of *notum* to generate a Wnt-low environment for head regeneration at anterior-facing wounds [30]. Follistatin is a broadly conserved negative regulator of Activin/Myostatin signaling ligands [31–34], and the *follistatin* RNAi head regeneration failure phenotype requires Activin signaling [27,28,30]. More specifically, the failed missing tissue response [28] and increased wound-induced *wnt1* expression at wounds [30] in *follistatin* RNAi animals is thought to act through modulation of Activin signalling. Whereas *follistatin* RNAi causes upregulation of Activin signalling, the impact of downregulation of Activin is less well understood. *ActR-1* (encoding an Activin receptor) RNAi can lead to ectopic pharynges [27] and some blastema splitting [35]. Activin and Follistatin have roles at wounds in multiple vertebrates [36–39]. These observations raise the possibility that Activin signaling is broadly utilized in animal regeneration, a possibility that remains poorly explored. Planarians are an attractive system to continue to dissect the mechanisms by which Activin signaling regulates regeneration.

Here we report an unexpected role for Activin signaling in controlling head-versus-tail regeneration through the regulation of asymmetric wound-induced *notum* expression at wounds in planarians. In animals with inhibited Activin signaling, ectopic posterior-facing heads form specifically during regeneration and are associated with a loss of asymmetry in wound-induced *notum* expression. We conclude that Activin has an essential role in the asymmetric activation of *notum* and in determining the head-tail regeneration decision at transverse amputation planes during planarian whole-body regeneration.

## Results

### *activin-2* is required for regeneration polarity and regeneration of AP-axis pattern

Planarians, as Platyhelminthes, are part of the Spiralia superphylum [40]. Genes from various spiralian species encode TGF-β-family signaling molecules, including planarian *activin-1* and *activin-2* [27,28,36]. Prior work indicates that some of these spiralian proteins belong to a TGF-β clade that includes vertebrate Activin and Myostatin [36]. Phylogenetic analysis places *Schmidtea mediterannea* Activin-2 in a clade primarily made up of Activins; however, whether this gene and other spiralian Activins are derived from an ancestral *myostatin* or *activin* gene

remains unknown (S1 Fig and S1 Data). Activin and Myostatin can signal through the same TGF-β receptors [41–43] and are sister groups within the TGF-β superfamily [44,45].

We inhibited the *S. mediterranea activin-2* gene by feeding dsRNA for RNA interference (RNAi). RNAi of *activin-2* (known as *activin* in [27]) was previously shown to suppress the *follistatin* failed regeneration RNAi phenotype [27,28]. After differing times of gene inhibition with RNAi, some animals were amputated and analyzed at time points post amputation (Fig 1A). *activin-2* RNAi (S2A Fig) resulted in a novel phenotype involving both regeneration of a head at posterior-facing wounds and AP-axis bifurcations in regenerating blastemas, leading, for example, to two heads in a single anterior blastema (Fig 1B). The combined presence of these two defects resulted in variable numbers of heads and tails in fragments with both head and tail amputated (trunk fragments) (Figs 1B and S2B). At anterior-facing wounds, only heads appeared. By contrast, at posterior-facing wounds, animals regenerated a presumptive tail or combinations of heads and tails (Figs 1B and S2B).

Ectopic posterior-facing heads in *activin-2* RNAi animals contained correctly positioned head anatomy, anterior head-tip (*sFRP-1*) and midline (*slit*) PCG expression domains, and developed a mid-body pharynx with reversed polarity, indicating that fully patterned anterior body axes developed with reversed polarity (Fig 1C). *activin-2* RNAi animals that did not develop ectopic heads displayed normal anatomy and PCG expression in regeneration, with

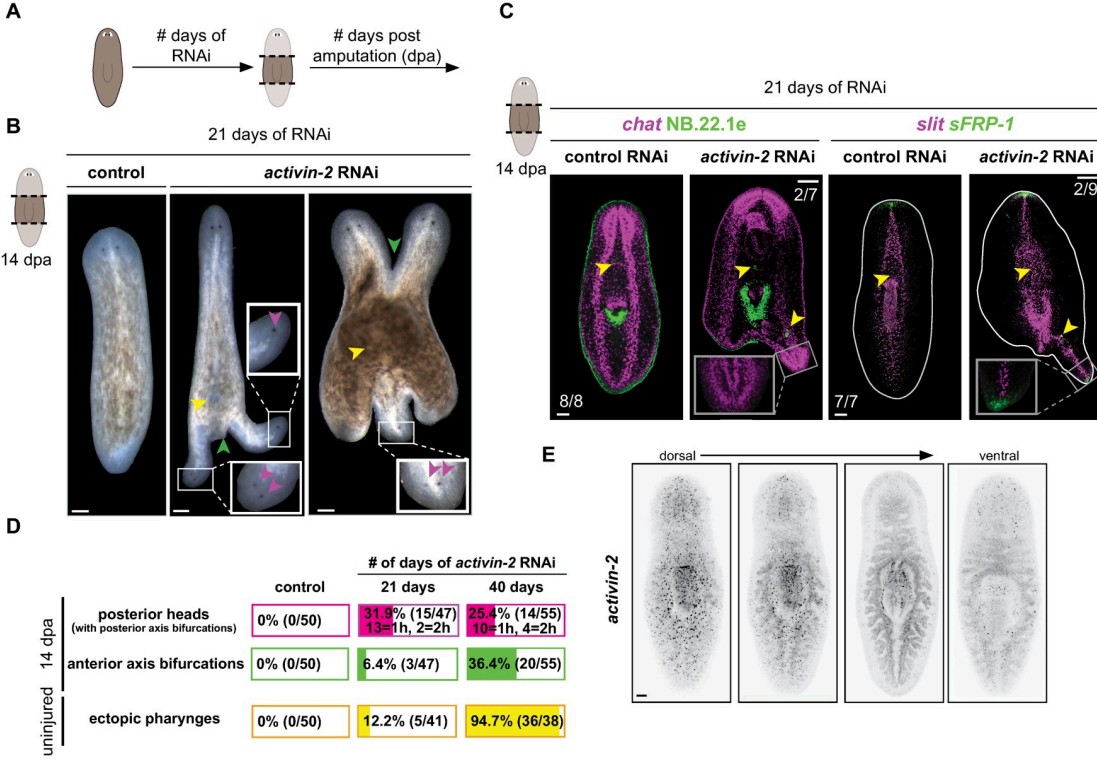

**Fig 1. *activin-2* is required for regeneration polarity and regeneration of AP axis pattern.** (A) Schematic of RNAi experiments where the number of days on RNAi and number of days post amputation (dpa) are specified separately. (B) Regenerating *activin-2* RNAi animals. Insets show magnified ectopic heads and eyes (magenta arrows). Green arrows, AP axis bifurcations; yellow arrows, ectopic pharynges. (C) FISH shows (left) CNS (*chat+*) and mouth and esophagus (*NB.22.1e*) and (right) PCG expression (midline *slit+*; anterior *sFRP-1+*) in ectopic axes. Inset: magnified view shows ectopic anterior. Yellow arrows: esophagi. (D) Quantification of posterior heads (based on presence of eyes) and axis bifurcations in regenerating animals, and ectopic pharynges seen in uninjured animals (DAPI) at 21 and 40 days after first RNAi. h is the number of posterior heads in a given animal. (E) *activin-2* expression pattern by FISH, from dorsal to ventral. All images are anterior up. Scale bars, 200 μm. Numbers of representative animals are indicated on bottom left of each panel.

the exception of ectopic posterior mouth tissue in some animals (S2C Fig). Trunk fragments were generated at early (21 days) and late (40 days) amputation time points after RNAi initiation, and were assessed for components of the *activin-2* RNAi phenotype after an additional 14 days post amputation. Polarity reversals were seen at the early regeneration time point, with penetrance not increasing with further time of RNAi (Fig 1D). Axis bifurcations at 21 days of RNAi were present mainly in the posterior and were associated with posterior-facing heads. Anterior axis bifurcation increased in penetrance over time–from 3/47 at 21 days RNAi to 20/55 at 40 days of RNAi (Fig 1D).

Both uninjured and regenerating *activin-2* RNAi animals displayed ectopic pharynges, mouths, and esophagi (Figs 1C and S2C–S2F). Animals were assessed for ectopic pharynges prior to amputation. At the early RNAi time point (21 days) 5/41 animals had an ectopic pharynx; at the late RNAi timepoint (40 days) 36/38 animals had an ectopic pharynx (Figs 1D and S2C–S2F). Ectopic pharynges [27] and anterior head splitting [35] have been observed in Activin receptor (ActR1, dd_3426) RNAi animals, consistent with a role for Activin signaling in regulating pharynx number and preventing axis splitting. Inhibition of other patterning genes, such as *ndl-3*, *wntP-2*, and *ptk7* can cause the formation of ectopic pharynges without causing axis bifurcations or polarity reversal, indicating that ectopic pharynx formation can be separable from the latter processes [15,16]. These observations–earlier abundance of polarity reversal than pharynx duplication in *activin-2* RNAi animals and the lack of polarity reversals in other RNAi phenotypes with ectopic pharynges–suggest that regeneration polarity reversal was not caused by the appearance of ectopic pharynges.

In other organisms, *activin* genes are expressed broadly in muscle, intestine, and at wounds [37,46–48]. This broad expression pattern allows Activins to act as humoral agents signaling environmental states. In *S. mediterranea*, *activin-2* was expressed throughout the planarian AP axis and most strongly medially (Fig 1E). This pattern stayed consistent after injury, 24 hours post amputation (S2G Fig). The medial enrichment of *activin-2* expression was previously shown in a regeneration time-course of *activin-2* expression [28]. *activin-2* was lowly expressed in currently available whole-animal scRNA-seq datasets (S2H Fig) [49]. Muscle-specific scRNA-seq data indicates that *activin-2* is expressed in *nkx1.1*$^+$ circular muscle cells, as well as a cluster described as DV-like but not well characterized (S2H Fig) [50]. Expression in body wall and pharyngeal muscle, and intestine, was confirmed with FISH (S2I Fig).

## *activin-2* RNAi results in regeneration-specific AP-polarity patterning defects

Many planarian RNAi phenotypes that affect patterning in regeneration also affect patterning in uninjured animals undergoing tissue turnover. Uninjured *activin-2* RNAi animals displayed mid-body widening associated with ectopic mouth cells, esophagi, and pharynges (Figs 2A and S3A). The aberrations to AP-axis polarity (posterior-facing heads) and AP-axis organization (duplicated heads/tails) that occurred during regeneration, however, did not occur during tissue turnover in *activin-2* RNAi animals (Figs 2A–2C and S3B). This raises the possibility that the AP-axis-patterning defects of *activin-2* RNAi animals reflect a requirement for *activin-2* in some regeneration-specific (i.e., not essential or occurring in tissue turnover) processes.

In principle, *activin-2* could be required for normal expression of patterning molecules during tissue turnover, such that alteration to the AP-patterning state results in polarity defects in regeneration. We therefore sought to determine whether homeostatic PCG expression region loss or reversal might occur in *activin-2* RNAi animals at 21 days of *activin-2* RNAi, when regeneration polarity was affected but regeneration axis splitting and pharyngeal duplication were minimally present. Transcripts for patterning genes that are normally expressed in the

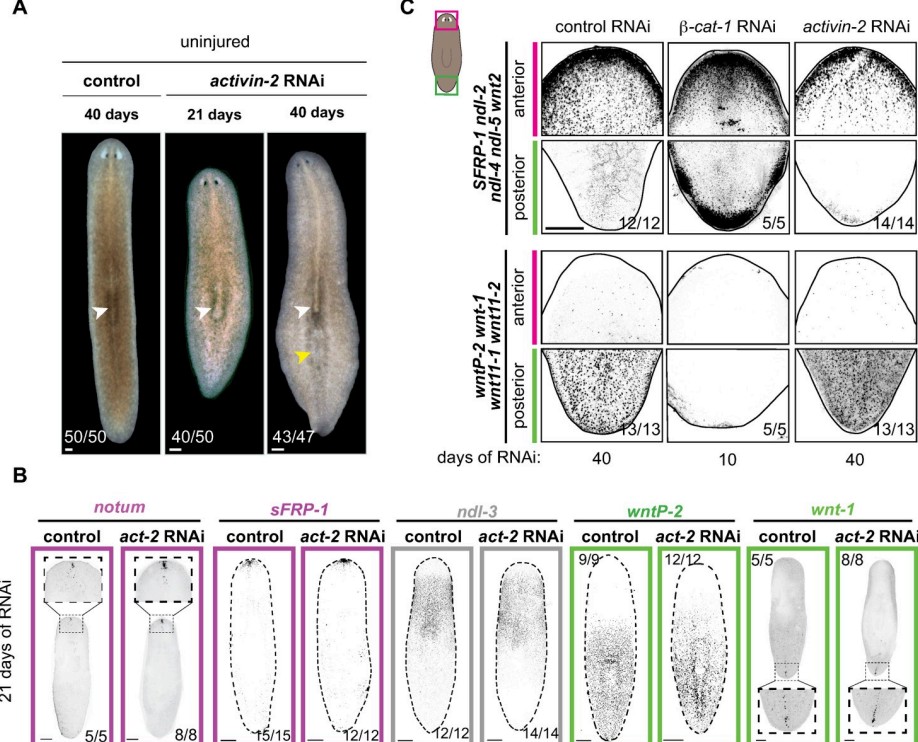

**Fig 2. Intact *activin-2* RNAi animals do not show polarity defects in patterning gene expression.** (A) Uninjured *activin-2* RNAi animals at 21 and 40 days after first RNAi feeding. Original pharynges (white arrows) and ectopic pharynx (yellow arrow) are shown. (B) FISH shows normal PCG expression and pole presence in uninjured *activin-2* (*act-2*) RNAi animals at 21 days after first RNAi feeding. (C) Intact *activin-2* RNAi animals display maintenance of anterior and posterior identity at 40 days RNAi by FISH. Bn contrast, *β-catenin-1* RNAi promotes a loss of posterior identity, and ectopic anterior identity in intact animals by 10 days of RNAi. (Top) A pool of anterior restricted genes (*sFRP-1*, *ndl-2*, *ndl-4*, *ndl-5*, and *wnt2*), (Bottom) A pool of posteriorly expressed genes (*wntP-2*, *wnt1*, *wnt11-1*, and *wnt11-2*). All images are anterior up. Scale bars, 200 μm.

anterior (*ndl-5*, *sFRP-1*), trunk (*ndl-3*, *wntP-2*, *ptk-7*), posterior (*wnt11-1*, *wnt11-2*, *and fz-4*), and poles (posterior *wnt1*, anterior *notum*) were present and in the proper relative domain order in *activin-2* RNAi animals (Figs 2B and S3B). We assessed boundaries and proportions of PCG expression domains, and found no significant changes, except for a subtle change in the anterior boundary of *ndl-3* expression (S3C Fig). After 40 days of *activin-2* RNAi, labeling with pools of anterior and posterior PCG RNA probes demonstrated that normal AP-axis polarization remained (Fig 2C).

These findings contrast with results for other genes known to control regeneration polarity outcomes, such as *β-catenin-1* [5–7] (Fig 2C). RNAi of *β-catenin-1* causes rapid and large-scale change of PCG expression domains throughout the body. *activin-2* RNAi animals were also assessed after 60 days of gene inhibition. Animals continued to widen, became less flat, and possessed large lateral bulges of tissue (S3G Fig). Tissue bulges lacked ectopic cephalic ganglia (S3H Fig). Despite body plan and muscle fiber organization changes, the relative expression positions of *ndl-3* and *wntP-2* expression were preserved (S3H Fig). Similarly, ectopic tissue bulges lacked ectopic expression of the anterior pole marker *notum* (S3I Fig). These findings suggest that Activin might control AP polarity using a mechanism that is distinct from constitutive Wnt pathway regulation and might be regeneration specific.

The midline also appeared normal at 21 days of *activin-2* RNAi. However, at 40 days of RNAi, a disorganized *slit*+ midline was apparent (S3D Fig). Disorganization of body-wall

muscle fibers was also seen only at this late time point, although body-wall muscle cell number and the overall dispersed pattern of the nuclei of these mononucleate cells was normal (S3E and S3F Fig). Midline disorganization was correlated temporally with the increase in anterior bifurcations during regeneration (Figs 1C and S3D).

## *activin-2* RNAi results in symmetric *notum* expression at wounds

The patterning defects (polarity and axis bifurcation) of *activin-2* RNAi animals observed only in regeneration prompted us to assess whether regeneration-specific adult processes required *activin-2*. We first assessed the process of wound induction of genes involved in re-setting PCG expression domains. FISH experiments demonstrated a robust loss of asymmetry of *notum* activation at 18h post-amputation in *activin-2* RNAi animals with high penetrance (95.4%) (Fig 3A). At this time point, there was not a significant difference in *notum* expression between *activin-2* RNAi and control anterior-facing wounds, or between *activin-2* RNAi anterior- and posterior-facing wounds (Fig 3B). This suggests that there was not a general overexpression of *notum* at wounds following *activin-2* RNAi, but a loss of *notum* inhibition at posterior-facing wounds. i.e., *notum* expression was no longer specific to anterior-facing wounds following *activin-2* RNAi.

To assess whether the change in *notum* expression reflected a defect in the wound response, affecting many genes, we utilized RNA sequencing. RNA sequencing also allows assessment of

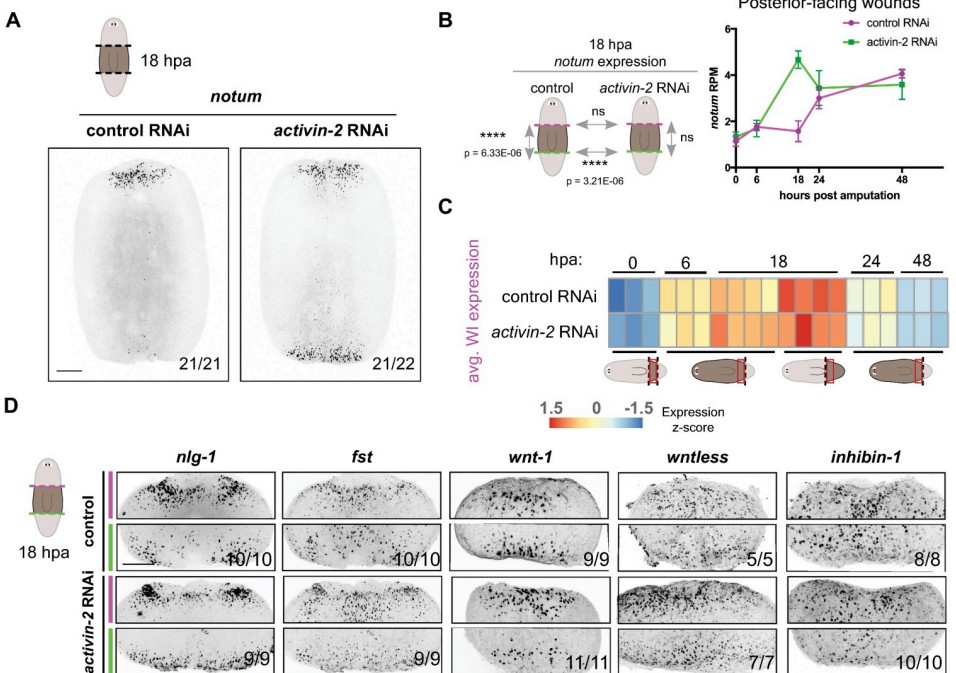

**Fig 3. *activin-2* is required for asymmetric activation of *notum* at anterior-versus-posterior wounds.** (A) FISH shows loss of wound-induced *notum* asymmetry (polarity) in *activin-2* RNAi animals at 18 hpa. (B) Cartoons show results from DESeq analyses: no significant difference between wound-induced *notum* expression at anterior- and posterior-facing wounds in *activin-2* RNAi animals, and increased *notum* expression at posterior-facing wounds in *activin-2* RNAi animals compared to controls at 18 hours post amputation. ns, not significant. Graph shows *notum* reads per million at posterior-facing wounds over time. (C) Heatmap shows expression of wound-induced genes as described [22] from bulk mRNA-seq of posterior-facing wounds at 0, 6, 18, 24, and 48 hours post amputation, and anterior-facing wounds at 18 hours post amputation. Cartoons show the wounds described in the heatmap. (D) FISH shows no difference in muscle wound-induced gene expression between control and *activin-2* RNAi animals. Magenta, anterior-facing wound; green, posterior-facing wound. All images are anterior up. Scale bars, 200 μm.

other events occurring in regeneration after the generic wound response. We performed RNA sequencing of posterior-facing wounds at 0, 6, 18, 24, and 48 hours post-amputation and of anterior-facing wounds at 18 hours post-amputation (S2 Data). We collected posterior-facing wounds at all time points because this is the site of ectopic *notum* expression, as well as anterior-facing wounds at 18 hpa to compare *notum* expression across wounds. No significant defect in muscle-specific gene expression was observed at the time of amputation (0 hours post-amputation) (S4A Fig). These data together with the data that muscle fibers and cell numbers were not overtly abnormal at this RNAi time point (S3E–S3F Fig) suggest that the *notum* expression phenotype was not caused by obvious changes to fiber number or morphology. No significant defect in the averaged expression of 128 planarian wound-induced genes occurred in *activin-2* RNAi samples (Figs 3C and S4B, and S3 Data). After the generic wound response phase, a missing tissue response occurs at major injuries and is associated with an enrichment of mitotic neoblasts at wounds, which can be detected as an increase in neoblast-specific transcripts at wounds in RNA-sequencing data. This occurred normally in *activin-2* RNAi animals (S4B Fig). These data indicate that the generic wound response and the missing tissue response both occurred in *activin-2* RNAi animals. Furthermore, *activin-2* expression itself was not detectably asymmetric at anterior- versus posterior- facing wounds both by FISH and by RNA sequencing (S4C Fig).

Although most assessed wound-induced genes were activated normally in *activin-2* RNAi animals, 6/128 displayed significantly different expression (adjusted $p < 0.001$) at 6 and/or 18 hpa in the RNA-sequencing data (S4 Data). *notum* was the most robustly changed gene in this data. We used FISH to examine the expression of some of these genes and observed that they were changed according to the sequencing experiment, validating our data (S4D Fig). We also performed FISH on other previously known planarian wound-induced genes, including a subset that are activated in muscle, and saw no overt differences in expression (Figs 3D and S4E). Among genes generically activated by wounding within hours of injury, *wnt1* promotes posterior identity [8,9]. However, wound-induced *wnt1* expression was not detectably aberrant in *activin-2* RNAi animals by FISH or RNA-seq (Fig 3D and S2 Data). Of all wound-induced genes assessed by RNA sequencing, *notum* displayed the highest $\log_2$(fold change) (1.51) and lowest adjusted p-value at the time point when AP-regeneration polarity defects emerged following *activin-2* RNAi (S4 Data).

Although not significantly different by RNA sequencing, *foxD* expression at wounds was higher in *activin-2* RNAi animals when compared with controls by FISH, but only at late time points following RNAi initiation (35 days)–temporally correlated with *slit*+-midline and animal widening (S4F Fig). *foxD* is wound-induced in *myoD*+ longitudinal muscle (S4G Fig) specifically at the ventral midline [23]; increased expression is therefore likely associated with midline widening.

### *activin-2* promotes the polarized response of longitudinal muscle fibers to wound orientation independent of AP location

Ectopic *notum* expression at posterior-facing wounds occurred by 18 hpa, but was not present at early time points. By 24 hpa normal wound-induced *notum* expression lowered in control and *activin-2* RNAi animals, reflecting the waning of wound induction (Fig 4A). Ectopic *notum* expression occurred at 21 days of *activin-2* RNAi and later by FISH (S5A Fig). We therefore used 18 hpa at 21 days of RNAi to further characterize ectopic *notum* at posterior-facing wounds. Some regeneration phenotypes associated with defects in regulation of Wnt signaling only affect head regeneration in fragments from the posterior (e.g., *follistatin* RNAi) [30]. We therefore tested whether the effects of *activin-2* RNAi on regeneration polarity are

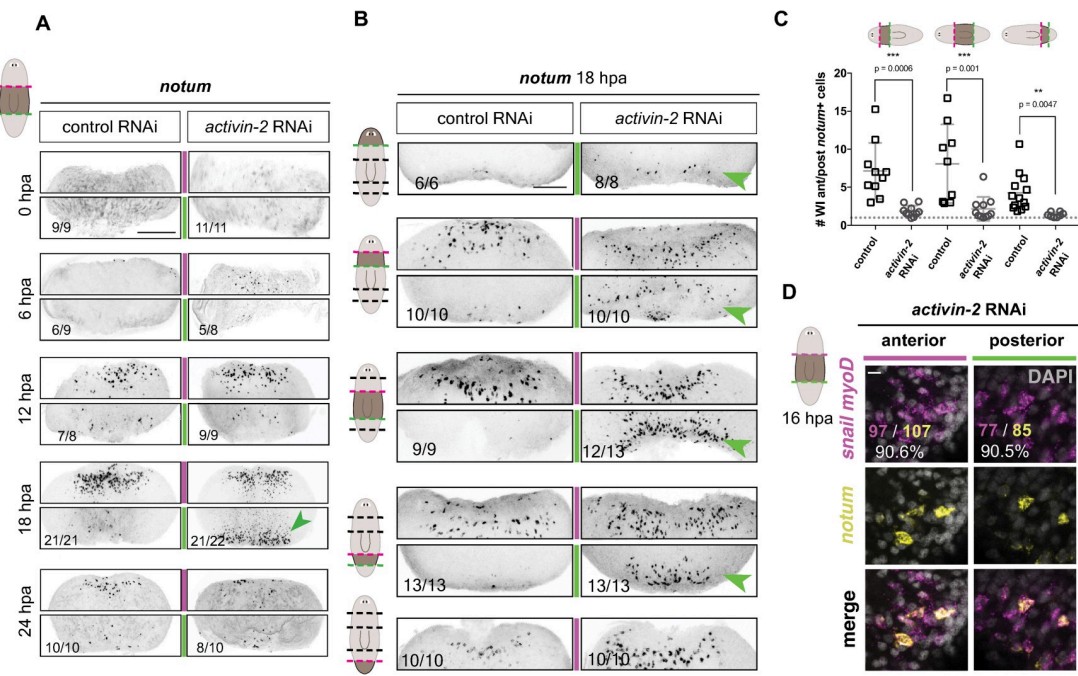

**Fig 4. *activin-2* promotes the polarized response of longitudinal muscle fibers to wound orientation independent of AP-axis position.** (A) Increased *notum* is seen at posterior-facing wounds specifically at 18 hours post amputation. (B) FISH shows increased *notum* expression at posterior-facing wounds across the AP axis. Animals were amputated into five fragments and assayed at 18 hours post amputation. (C) Graph shows quantification for (B). Data are shown as the ratio of *notum*+ cells found at anterior-facing versus posterior-facing wounds of a given piece. Kruskal-Wallis (one-way ANOVA on ranks) test with multiple comparisons was performed to determine *P* values. (D) FISH shows co-expression of wound-induced *notum* and *myoD, snail* at anterior- and posterior-facing wounds in *activin-2* RNAi animals. Number of *myoD/snail*+ cells that co-localize with *notum* (magenta) out of total *notum*+ cells (yellow). White number, percentage of *notum*+ cells that are longitudinal fibers. All images are anterior up. Scale bars, 200 μm, except for high magnification panels in where they represent 10 μm.

region specific. Elevated wound-induced *notum* expression at posterior-facing wounds occurred at every AP-amputation location examined, suggesting *activin-2* RNAi impacted regeneration polarity across the entire AP axis (Fig 4B and 4C).

Wound-induced *notum* expression at anterior-facing wounds in wild-type animals occurs specifically in longitudinal muscle cells, which are oriented along the AP axis and express *myoD* and *snail* [22]. Ectopic *notum* expression at posterior-facing wounds of *activin-2* RNAi animals was similarly specific to *myoD*+/*snail*+ cells (Fig 4D). These findings are consistent with the possibility that biology intrinsic to longitudinal muscle is responsible for generating *notum*-expression polarity at wounds, and that *activin-2* is required for this process.

### *activin-2* promotes the local polarized response of newly specified longitudinal muscle fibers to wound orientation

Because the loss of *notum* polarity in *activin-2* RNAi animals was observed at 21 but not at 7 and 14 days post-RNAi initiation, we considered the possibility that *activin-2* was required during muscle cell turnover to maintain regeneration polarity. In this scenario, it would take time for a substantial number muscle cells at posterior-facing wounds to have been generated in cell turnover under *activin-2* RNAi conditions to observe ectopic *notum* expression. *activin-2* transcripts were significantly reduced by 7 days RNAi by qPCR (S5B Fig). Because neoblasts are the only cycling somatic planarian cells [51–54], EdU labeling marks a subset of newly formed cells originating from neoblasts. Consistent with the possibility that *activin-2* RNAi

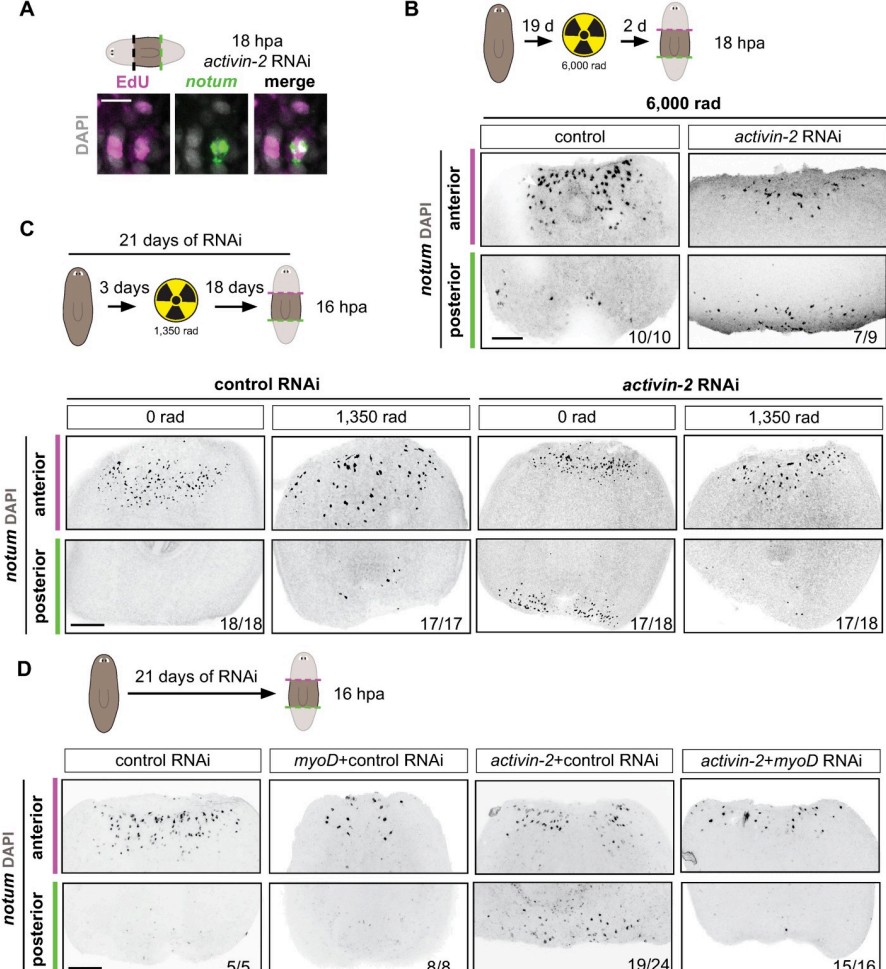

**Fig 5.** *activin-2* **promotes the polarized response of newly specified longitudinal muscle fibers to wound orientation.** (A) EdU incorporation in *notum+* cells at posterior-facing wounds in an *activin-2* RNAi animal. Lower magnification in S5A. (B) Top: cartoon shows the experimental design. Bottom: FISH shows *notum* expression at 18 hours post-amputation 21 days after the first RNAi feeding in lethally irradiated (6,000 rads) animals. (C) Top: cartoon shows the experimental design. Bottom: FISH shows *notum* expression at 18 hours post-amputation 21 days after the first RNAi feeding where a sublethal dose of irradiation (1,350 rads) was given 19 days prior to amputation. (D) Top: Cartoon showing experimental design. Animals were fed every 3 days. Animals were fed either control or *activin-2* dsRNA for feeding 1, 2, 3, 5, and either *myoD* or control for feeding 4. Bottom: FISH shows *notum* expression at 18 hours post-amputation 21 days after the first RNAi feeding. Magenta, anterior-facing wounds; green, posterior-facing wounds. All images are anterior up. Scale bars, 200 μm, except for high magnification panels in where they represent 10 μm.

affected the ability of newly made longitudinal muscle cells to display polarity, EdU labeling showed that *notum*$^+$ longitudinal fibers at anterior- and posterior-facing wounds in *activin-2* RNAi animals included newly generated cells (Figs 5A and S5C).

To further explore this possibility, we first utilized irradiation, which can reduce or eliminate neoblasts [55,56]. As a control, lethal irradiation (6,000 rads) two days prior to amputation eliminated neoblasts acutely in *activin-2* RNAi animals and wound-induced *notum* was still symmetric (Figs 5B and S5D). Neoblasts are therefore not acutely required for wound-induced *notum* expression at *activin-2* RNAi posterior-facing wounds. Sublethal irradiation causes a reduction in neoblasts, which was used to decrease cell turnover while allowing animals to survive for a long enough period of time to observe the impacts of reduced cell

turnover. Animals were sublethally irradiated (1,350 rads) after 3 days of RNAi to severely reduce but not eliminate neoblasts, and amputated at 21 days of RNAi. These *activin-2* RNAi animals displayed a marked decrease in posterior-facing *notum* expression compared to unirradiated controls or to lethally irradiated (6,000 rads) animals (Figs 5C and S5D and S5E). This is consistent with the possibility that irradiation blocked the formation of new muscle cells in *activin-2* RNAi animals and that it is the newly differentiated muscle cells that display ectopic *notum* expression at posterior-facing wounds.

We also tested whether muscle cell turnover is required for the loss of *notum* asymmetry in *activin-2* RNAi animals by using RNAi of *myoD*. *notum* is wound-induced in longitudinal muscle cells and *myoD* is required for new longitudinal muscle cell production in tissue turnover [22]. Therefore, RNAi of *myoD* causes a gradual decline in longitudinal muscle because new muscle cells fail to form and replace dying muscle cells during tissue turnover. Concurrent inhibition of *myoD* and *activin-2* RNAi, to inhibit new muscle production, robustly blocked ectopic *notum* expression at posterior-facing wounds (Fig 5D); i.e., *notum* expression asymmetry (stronger at anterior-facing wounds) was present despite *activin-2* RNAi. In this experiment, wound-induced *notum* expression was also decreased but not gone at anterior-facing wounds, as expected. There were similar levels of *activin-2* transcript reduction between conditions (S5F Fig). The suppression of the *activin-2* RNAi phenotype with irradiation or *myoD* RNAi further suggests that the *activin-2* phenotype involves defects in the ability of newly generated longitudinal muscle cells to display asymmetry in *notum* activation at wounds.

## Anterior axis splitting in *activin-2* RNAi animals is associated with the formation of two anterior poles

By 48-72h post-amputation, the anterior pole forms at the pre-existing midline where it acts as a signaling center to influence ML and AP pattern in the blastema [23,24,26,57,58]. This time of regeneration is subsequent to the generic wound-response phase when *notum* is activated at anterior-facing wounds in pre-existing muscle cells. The regenerating anterior and posterior poles are newly generated cells with fates specified to make pole cells by transcription factor expression (e.g., *foxD* for the anterior pole) in neoblasts [23–25].

At anterior-facing wounds, some *activin-2* RNAi animals displayed the nucleation of split anterior poles at 72 hpa (Fig 6A). Given the organizer-like role of anterior poles in promoting midline and AP pattern, this suggests split anterior pole nucleation resulted in double heads in anterior *activin-2* blastemas. This split pole defect, associated with head duplication in anterior blastemas, occurred primarily at the late RNAi time point (40 days) when animals displayed multiple body shape abnormalities including widening. Anterior pole cell numbers (*notum*+ and/or *foxD*+) were similar between control and *activin-2* RNAi uninjured animals at 21 or 40 days on RNAi, as well as in 72 hpa blastemas (S6A and S6B Fig). Some poles were not split but wider than in controls after 40 days of RNAi (S6B and S6C Fig).

Head blastema and anterior pole bifurcation is similar to the defect caused by RNAi of *nkx-1.1*, which is required for the maintenance of ML axis-oriented circular body wall muscle fibers [22]. *activin-2* is expressed in circular muscle fibers, and its expression therefore decreases with circular fiber loss [22]. This suggests that the blastema bifurcation phenotype of *nkx1.1* RNAi animals might be associated with reduced expression of *activin-2*. Furthermore, there was elevated wound-induced *notum* expression at the posterior-facing wounds of head fragments in *nkx1.1* RNAi animals (S6D Fig). During early regeneration (48 hpa) muscle was present at *activin-2* RNAi wound sites (S6E Fig) and *activin-2* RNAi animals contracted wounds to promote wound closure normally (S6F Fig).

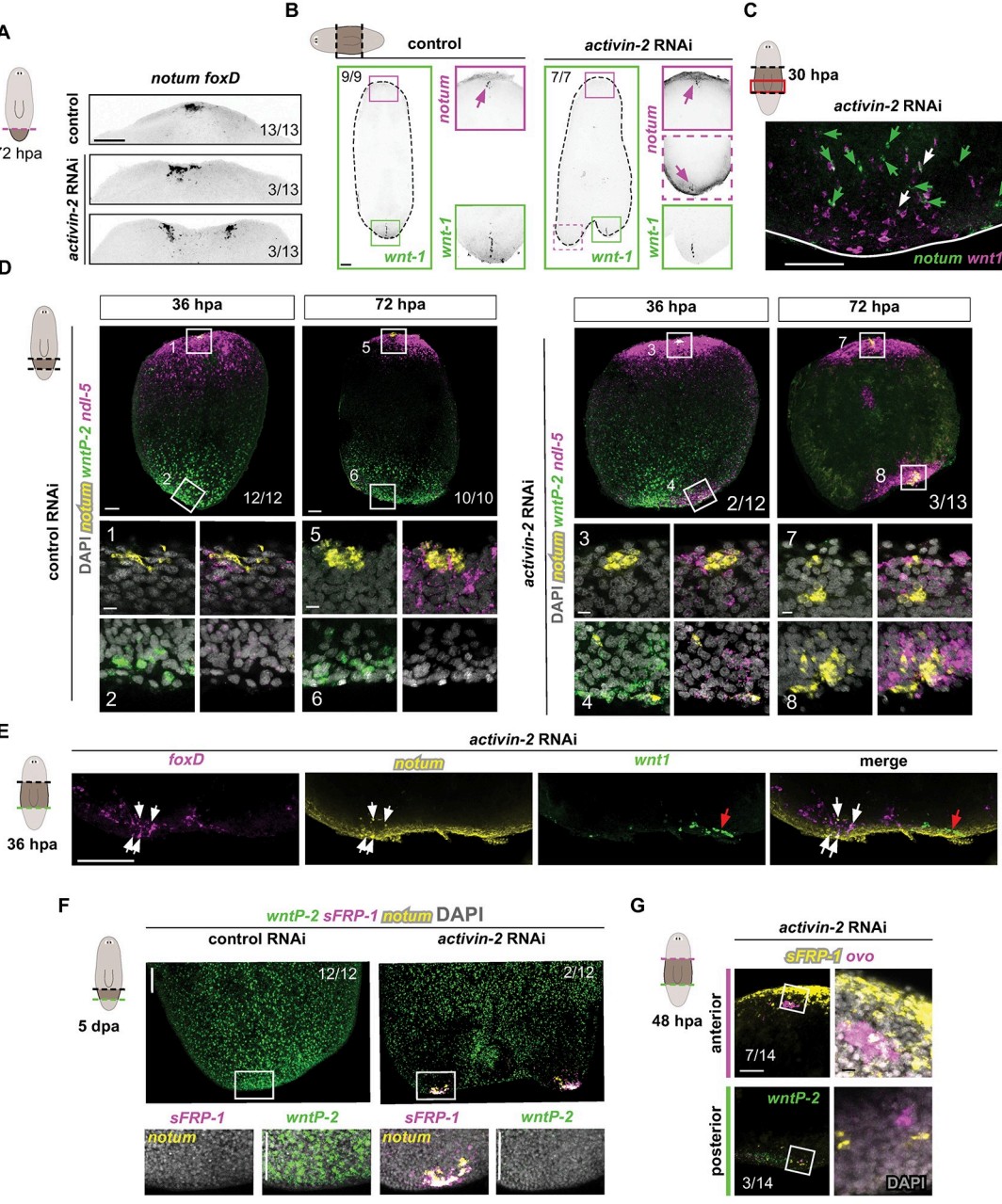

**Fig 6. Ectopic wound-induced *notum* is correlated with the formation of an ectopic anterior pole in a Wnt-high environment.** (A) FISH shows split anterior poles (*notum/ foxD+*) in 72 hours regenerating *activin-2* RNAi animals. (B) FISH show anterior (*notum*) and posterior (*wnt1*) pole markers at 12 days post amputation in *activin-2* RNAi animals. (C) FISH shows expression of *notum* and *wnt1* in the same posterior-facing wound of an a*ctivin-2* RNAi regenerating fragment at 30 hours post amputation. Green arrows show *notum+* cells. White arrows denote cells that co-express *notum* and *wnt1*. (D) FISH shows regenerating fragments resetting PCG (*ndl-5*, *notum*, *wntP-2*) expression and nucleating new anterior poles at different time points. *activin-2* RNAi animals show ectopic anterior poles at posterior-facing wounds. Boxes indicate high magnification images shown below. (E) FISH shows anterior pole markers (*notum* and *foxD*), and the posterior pole marker (*wnt1*) at the same posterior-facing wound of a regenerating fragment at 36 hours of regeneration. Zoom out seen in S7A Fig. (F) FISH shows posterior-facing wound of regenerating fragments at five days. Posterior related (*wntP-2*) and anterior related (*notum/sFRP-1+*) gene expression by FISH. (G) FISH shows *ovo+* eye progenitors and expression of anterior (*sFRP-1*) and posterior (*wntP-2*) PCGs at posterior-facing wounds of an *activin-2* RNAi animal. Boxes: zoom ins on the right. Magenta, anterior-facing wound; green, posterior-facing wound. Scale bars, 200 μm, except for high magnification panels in where they represent 10 μm.

## Ectopic wound-induced *notum* is correlated with the formation of an ectopic anterior pole in a Wnt-high environment

Posterior-facing wounds in *activin-2* RNAi animals always regenerated at least one posterior pole, but also displayed simultaneous production of discrete anterior and posterior poles (Fig 6B). At 30 hpa, *notum*[+] and *wnt1*[+] cells were present in variable, but intermingled distributions at posterior-facing wounds; at this time point positive cells could include both pole progenitors and cells with residual wound-induced expression. Foci of cells reflecting new poles were not yet present (Figs 6C and S7A). Despite the largely dispersed and intermingled pattern of cells expressing these genes at this early time point, two distinct foci of either anterior or posterior pole cells formed later in regeneration.

By 36–38 hpa the anterior PCGs *ndl-5* and *sFRP-1* were ectopically expressed in a locally clustered manner at posterior-facing wounds, prior to substantial ectopic anterior-pole coalescence (Figs 6D and S7B). *notum*[+] cells were present at this time, and localized to the regions of *ndl-5* and *sFRP-1* expression. However, there were few *notum*[+] cells and they were not yet coalesced into tight foci reflecting new poles. At this time (38 hpa), posterior PCG expression (*wntP*-2) was still broad at the wound, but was reduced in level locally in the region of ectopic anterior PCG expression clusters (Figs 6D and S7B). The *notum*[+] cells at 36 hpa were *foxD*[+], indicating that they were pole progenitors and/or pole cells (Fig 6E). *wnt1*[+] cells at this time were also regional and no longer intermingled with *notum*[+] cells. Instead, local and separate locations of *notum*[+] and *wnt1*[+] cells at the same wound face were emerging (Fig 6E). In summary, by around 36 hpa ectopic local anterior PCG expression in *activin-2* RNAi animals was associated with local reduction in *wntP-2* expression and early stages of ectopic anterior pole formation spatially separated from posterior pole cells.

By 48 hpa both *notum*[+]; *foxD*[+] anterior poles and *wnt1*[+] posterior poles showed increased emergence at different locations (S7C Fig). By 72 hpa, all *notum*[+] ectopic anterior-pole cells at posterior-facing wounds had coalesced and the ectopic *ndl-5*[+] regions were expanded and stronger (Fig 6D). At 5 dpa, regenerating animals still possessed a global posterior *wntP-2*[+] zone, with *wntP-2* expression being only locally cleared near anteriorized regions (Fig 6F). Reduction of *wntP-2* expression near anterior PCG foci was stronger at 5 dpa than when initial anterior PCG expression was detected at 36 hpa. In bulk RNA-sequencing data from *activin-2* RNAi animals, an increase in posterior identity at posterior-facing wounds occurred normally, consistent with the local nature of the phenotype and its incomplete penetrance (S7D Fig). These molecular analyses only identified three combinations of heads and tails at posterior-facing wounds in *activin-2* RNAi animals: tail only, tail-head, and head-tail-head.

These findings indicate the local nature of the patterning changes that occurred at *activin-2* RNAi wounds–with local anterior PCG expression in part of the blastema associated with locally reduced posterior PCG expression, and with ectopic anterior pole formation only occurring near the blastema region with focal anterior PCG expression. This explains how both heads and tails were able to regenerate from single posterior-facing wounds in *activin-2* RNAi animals. Furthermore, the percentage of *activin-2* RNAi animals with *ndl-5*[+] and *notum*[+] foci at posterior-facing wounds at 72 hpa (23.1%) (Fig 6D), or ectopic *foxD*[+]/*notum*[+] poles (18.2%) (S7E Fig), was similar to that of animals with ectopic posterior heads (31.9%) (Fig 1D). This suggests that ectopic wound-induced *notum* might only result in posterior-facing heads when ectopic anterior PCG induction occurred and was associated with reduced local Wnt activity and anterior pole formation.

PCGs are hypothesized to influence the specialization choices of neoblasts [3]. For example, RNAi of *bmp4* results in dorsal epidermis-specialized neoblasts expressing ventral markers [59]. Consistent with this hypothesis, posterior-facing wounds in *activin-2* RNAi animals had

local eye-specialized neoblasts (*ovo*+) at 48 hours post-injury (Figs 6G and S7F). The location of these progenitors was proximal to anterior-PCG *sFRP-1*+ cells, suggesting that local changes in Wnt signaling led to local alteration in the fate choices of neighbouring neoblasts and the regeneration of posterior-facing head cell types that could organize into an ectopic posterior head.

## Discussion

Pattern formation is the process of specifying the identity and organization of cells in spatial arrangements. Patterning in regeneration initiates at wound faces that are unpredictable in location and shape, involves production of variable combinations of missing cell types, and involves patterning of new tissue in the context of pre-existing mature tissues. These unique challenges suggest the existence of regeneration-specific patterning mechanisms. The resetting of the expression domains of patterning genes in muscle after injury has been proposed to drive planarian regeneration [3,22,30]. How patterning gene expression domains are reset after injury is poorly understood, but involves wound signaling. *notum* encodes a Wnt antagonist and is unique among known wound-induced genes in being preferentially activated at anterior- over posterior-facing wounds, where it promotes head identity [12]. How the polarity of wound-induced *notum* is established has remained unknown, but occurs in longitudinal (AP axis-oriented) muscle fibers [22]. One model is that some prior property of adult tissue, "polarity", is a pre-existing architectural cue leveraged by wounds to trigger asymmetric outcomes for anterior-facing and posterior-facing blastemas.

Prior work on regeneration polarity has identified components of Wnt signaling as required for the head-versus-tail regeneration outcome [5–7,9,10,12]. However, in *APC* RNAi animals, which have upregulated Wnt signaling [6], the asymmetry of *notum* activation at anterior-facing over posterior-facing wounds remains [12]. Wnt signaling itself might therefore not be a direct regulator of the polarity mechanism that enables asymmetric *notum* activation at wounds. Similarly, Hedgehog signaling is known to impact the head-versus-tail regeneration decision in planarians [60, 61]. *hedgehog* RNAi animals fail to regenerate tails [60–62] and RNAi of *patched*, which causes upregulation of the Hedgehog pathway, causes regeneration of tails in place of heads [60,61]. However, *patched* RNAi does not impact the asymmetric activation of *notum* at wounds [12]. Therefore, no signaling pathway is as yet established to be required for this upstream most polarity step in the process of head regeneration. Here, we found that Activin signaling is required for AP regeneration polarity, involving a requirement for wound-induced *notum* asymmetry. *activin-2* inhibition had no detectable impact on the ability of homeostatic Wnt signaling to maintain AP PCG expression, consistent with the possibility that Activin impacts some process required for regeneration polarity but not homeostatic maintenance of the order of existing PCG expression patterns. Our data indicate that *activin-2* affects the ability of newly specified longitudinal muscle cells to be regeneration polarity competent. Perturbing muscle production, such as with *myoD* RNAi, does not cause loss of *notum* expression polarity in remaining fibers; furthermore, longitudinal fiber number was normal in *activin-2* RNAi animals with polarity defects. Therefore, the requirement for *activin-2* in regeneration polarity is not simply explained by a defect in longitudinal muscle production. One hypothesis is that longitudinal muscle cells have AP polarity and that this orientation impacts the asymmetry of wound-induced *notum* expression. Inhibition of Activin could perturb the polarization of newly synthesized longitudinal muscle or a different process essential for the capacity of longitudinal muscle to properly regulate *notum* activation asymmetrically. Future work could aim to understand the relation between *activin-2* signaling and asymmetric processes that result in longitudinal muscle displaying preferential activation of *notum* at anterior over posterior-facing wounds.

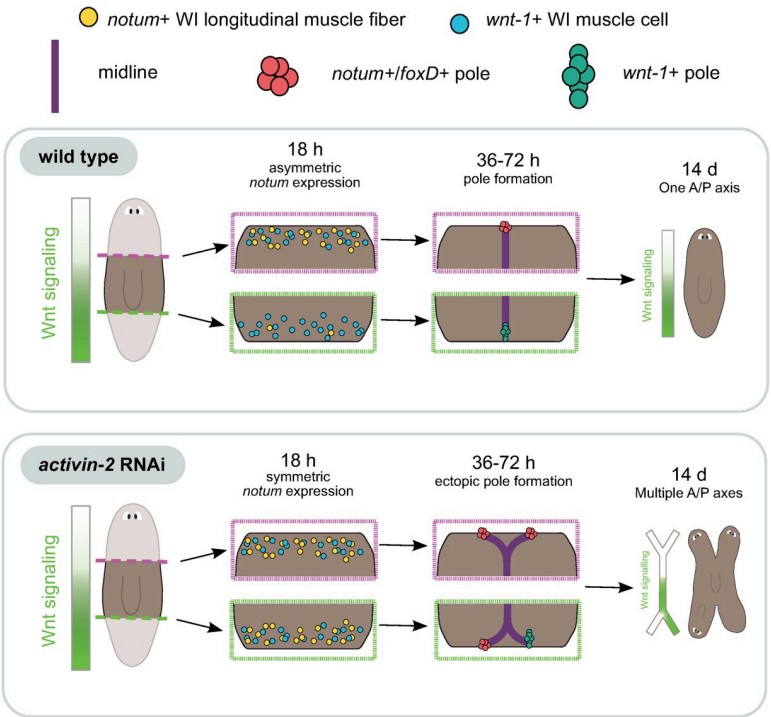

**Fig 7. *activin-2* RNAi animals display a regeneration-specific polarity defect in patterning correlated with a loss of wound-induced *notum* polarity.** Model of regeneration with *activin-2* RNAi defects depicted. Animals possess a *wnt* expression gradient from posterior to anterior. Upon amputation in *activin-2* RNAi animals, *notum* wound-induced gene expression (6–24 hr) is no longer spatially restricted to the anterior. Multiple midlines and poles can form, including anterior poles at the posterior-facing wound (36-72 hpa). Posterior identity is maintained in regenerated animals, with ectopic anterior axes (14 d).

Ectopic *notum* expression at posterior-facing wounds in *activin-2* RNAi animals was associated with ectopic anterior PCG expression, local reduction of posterior *wnt* expression, and ectopic anterior pole formation (Fig 7). Unexpectedly, this phenomenon could result in the simultaneous formation of discrete anterior and posterior poles and a head and a tail emerging from individual posterior-facing wounds. We propose that ectopic wound-induced *notum* expression can result in some anterior PCG activation, and that this can in some but not all cases lead to tipping points of local stable anterior identity. Why stable anterior PCG activation appears localized to only a region of the posterior-facing blastema and in only some animals is not fully understood. However, one possibility is that there is variability in how much anterior PCG activation is caused by ectopic *notum* expression at posterior-facing wounds. Local tipping points could then be stochastically reached, with sufficient anterior PCG expression resulting in more anterior PCG expression in new muscle cells. Such a self-reinforcing process could ultimately be stabilized by formation of an anterior pole from neoblasts choosing an anterior pole fate near anterior PCG expression foci. With a nucleated anterior pole, local posterior PCG expression inhibition and stable anterior PCG expression would occur. In many cases, a tipping point would not be reached, with posterior PCG expression dominating the entire wound and no ectopic anterior pole(s) forming. *notum* does not turn on as early at *activin-2* RNAi posterior-facing wounds as it does at wild-type anterior-facing wounds, possibly explaining why this process is less robust than wild-type head formation. There could also be additional mechanisms that distinguish anterior- and posterior-facing wounds. Regardless, in essentially all cases involving ectopic anterior pole formation at posterior-facing wounds in

*activin-2* RNAi animals, posterior pole formation also occurred–anterior identity foci formation might have only been compatible with posterior pole formation when it was sufficiently spatially separated from it.

At anterior-facing wounds, *activin-2* was required for the pattern of nucleation of the anterior pole, with inhibition of *activin-2* causing the occasional nucleation of two poles. The regeneration-specific head-splitting aspect of the *activin-2* RNAi phenotype was temporally correlated with midline widening and nucleation of multiple poles. These processes requiring *activin-2* –wound-induced *notum* activation and the *de novo* nucleation of poles–are two processes unique to regeneration as compared to tissue turnover (Fig 7).

Activin signaling has been implicated in several processes in planarian biology from prior work. Inhibition of planarian *follistatin*, which encodes a conserved Activin inhibitor, can cause head regeneration failure when amputation occurs in the posterior [27,28,30]. This is associated with regulation of the levels of wound-induced *wnt1*, and *activin-1* RNAi can suppress this phenotype [30]. Furthermore, *follistatin* RNAi causes a defect in the missing tissue response and this defect is also suppressed by RNAi of either *activin-1* or *activin-2* [28]. *follistatin* is activated by wound signaling [27,28,30], and these findings implicate inhibition of Activin by Follistatin in setting the levels of wound-induced *wnt1* and in promoting the missing tissue response. *notum* expression was not determined to be aberrant in *follistatin* RNAi animals [30], unlike the case for *activin-2* RNAi animals. Our data suggests that *activin-2* does not regulate *notum* wound-induced levels *per se*, but specifically is required to prevent *notum* from being activated at posterior-facing wounds. *follistatin* RNAi should result in increased Activin-2 signaling and might therefore simply result in increased Activin-2-mediated promotion of polarity, with polarity therefore being normally generated. Inhibition of an Activin receptor-encoding gene can result in ectopic pharynges (*ActR-1*, dd_3426*)* [27] and anterior head splitting (ActR-1, dd_3426) [35]. However, Activin signaling was previously not known to regulate regeneration polarity at posterior-facing wounds. Together these findings highlight the prominent role of Activin in regulating processes at planarian wounds associated with key steps of regeneration.

Pattern formation in development involves prominent signaling pathways, including different Tgf-β signaling pathways. Tgf-β signaling ligands are subdivided into two major groups, which signal through different receptors and Smad-family proteins: (i) Bmp and (ii) a group including Activin, Myostatin, Tgf-β, and Nodal. Multiple of these signaling ligands have important roles in patterning embryos. For example, Bmp patterns the dorsal-ventral axis of embryos throughout the animal kingdom [63]. Bmp also patterns the adult planarian DV axis [64–66]. Among the other ligand group, Nodal has major roles in the specification and patterning of embryonic mesendoderm and in the development of left-right asymmetry [67–72]. By contrast, the Activin-Follistatin pathway has not been identified as a major regulator of patterning in animal development and instead is more prominently associated with non-patterning or adult processes [72,73]. In zebrafish heart and fin regeneration, the Activin pathway ligand-encoding gene *inhbaa* and the Activin receptor-encoding *alk4* gene are expressed at wounds and are required for proper regeneration [38,39]. In mice, *follistatin* and *activin* are expressed at dermal wounds [37,46] and have roles in repair [74,75]. In axolotls, a Follistatin-like molecule, *kazald-1* is specifically expressed in the adult regenerating limb and is required for regeneration [76]. Activin and the related Myostatin also have roles in adult homeostatic processes providing, for example, inter-organ humoral signals to regulate fat body and endocrine functions [47,48,77–79]. These findings raise the possibility that the Activin-Follistatin signaling pathway has broad roles in adult biology that are conserved in evolution, including in wound repair and regeneration. The many and distributed expression locations of *activin-2* in planarians are consistent with the possibility that Activin might, similar to the case in other

organisms, have a broad-acting (rather than only local) signaling role [73]. Our findings suggest some of these roles can include regeneration-specific pattern-initiating processes, and that Activin signaling might prove to be a major regulator of regeneration in animals. This will be an important possibility to explore in multiple additional regenerative organisms.

In conclusion, an Activin-family signaling ligand has essential roles in events occurring at wounds that initiate pattern in planarian regeneration. Disruption of these processes leads to body plan aberration with animals regenerating multiple heads and heads formed with reversed polarity. Activin is required for the earliest known step in establishing head-versus-tail regeneration–the polarized expression of *notum* that is selective for anterior-facing wounds. We suggest that Activin signaling is an important regulator of Wnt signaling at wounds to promote pattern in whole-body regeneration.

## Methods

### Double-strand RNA synthesis and RNAi

dsRNA was synthesized using *in vitro* transcription reactions (Promega) using PCR-generated templates with flanking T7 promoters (TAATACGACTCACTATAGGG). 16 ul of template reaction was mixed with 1.6 ul of each rNTP (100 mM); 0.6 ul dithiothreitol (1M DTT); 4 ul T7 polymerase; and 24 ul of 5x Transcription buffer. Reactions were incubated overnight at 37˚C. Forward and reverse reactions were then combined, followed by ethanol precipitation, resuspension in 30 ul of water and annealing (95˚C for 5 minutes, room temperature for 20 minutes). dsRNA concentration was between 5–8 ug/ul. Animals were fed twice per week using a ~2:1 ratio of homogenized calf liver to dsRNA (13 ul dsRNA, 26 ul of calf liver, and 1 ul of dye per aliquot). At each feeding excess food was added to each dish so that not all food was finished; approximately 4 ul per worm. In all cases, animals were fixed seven days after the last feeding. For regeneration and RNA-seq experiment, animals were amputated one week after the last RNAi feeding. Contraction experiments placed some amputated animals directly into 100% modified Holtfreter's solution (3.5 g/L NaCl, 0.2 g/L NaHCO$_3$, 0.05 g/L KCl, 0.2 g/L MgSO$_4$, 0.1 g/L CaCl$_2$, pH 7.0–7.5) to inhibit muscle movement as a negative control for wound contraction.

### Fixation

Animals were killed in 5% N-acetyl-cysteine (NAC) in PBS for 3 minutes before fixation in 4% formaldehyde in PBSTx (PBS + 0.3% Triton X-100) for 15 minutes. Fixative was removed and worms were rinsed with PBSTx. Animals were dehydrated and stored in methanol at -20˚C.

### Whole-mount fluorescence *in situ* hybridizations and immunostainings

RNA probes were synthesized using *in vitro* transcription reactions (Promega) and whole-mount FISH was performed as previously described (King and Newmark 2013), with minor modifications. Briefly, animals were rehydrated, bleached in a formamide based solution on a light table and treated with proteinase K (2 µg/ml). Following overnight hybridizations, samples were washed twice in each of pre-hybridization buffer, 1:1 pre-hybridization-2X SSC, 2X SSC, 0.2X SSC, PBS with Triton-X (PBST). Subsequently, blocking was performed in 0.5% Western Blocking Reagent (Roche, 11921673001) and 5% inactivated horse serum PBSTx solution and anti-DIG, anti-DNP, or anti-FITC antibodies were used overnight at 4˚C. Washes and tyramide development were as previously described. Peroxidase inactivation with 1% sodium azide was done for 90 minutes at room temperature. An anti-muscle mouse monoclonal antibody 6G10 was used in a 1:1,000 dilution, and an anti-mouse Alexa-488 conjugated

antibody (Life Tech) was used in a 1:500 dilution. Samples were stained with DAPI overnight (Sigma, 1 mg/ml in PBSTx). Nitroblue tetrazolium/5-bromo-4-chloro-3-indolyl phosphate (NBT/BCIP) colorimetric whole-mount in situ hybridizations (ISH) were performed as described[80].

## Imaging

Fluorescence images were taken with a Zeiss LSM700 Confocal Microscope using ZEN software or with a Leica SP8 Confocal Microscope. Image analysis was performed using Fiji/ImageJ. For each channel, histograms of fluorescence intensity were used to determine the cut-off between signal and background. All FISH images are representative of all images taken in each condition. Light images were taken with a Zeiss Discovery Microscope. Cell counting was performed manually after blinding control and experimental conditions.

## RNA-seq experiments

Total RNA was isolated using Trizol (Life Technologies) from animal fragments. Libraries were prepared using the Kapa Stranded mRNA-Seq Kit Illumina Platform (KapaBiosystems). Libraries were sequenced on an Illumina Hi-Seq. Libraries were mapped to the dd_Smed_v6 transcriptome (http://planmine.mpi-cbg.de); using bowtie v1.1.2 with -best alignment parameter. The number of mapped reads per contig in every cell was quantified using the coverageBed utility from the bedtools v2.26.0 suite and reads from the same isotig were summed to generate raw read counts for each transcript. Pairwise differential expression analysis was performed using DESeq2. Expression values from DESeq normalization were scaled, row-wise, to generate z-scores for heatmaps and visualized using the pheatmap package. Significance is reported as padj values, with padj < 0.05 used as a cutoff.

## Irradiation

Animals were irradiated using a dual Gammacell-40 137 cesium source to deliver either 1350 or 6,000 rads.

## EdU delivery and labeling

Animals were fed EdU (Sigma #T511293) diluted in liver to 0.5 mg/ml and 1% DMSO. Following feeding, animals were incubated in 5g/L Instant Ocean until fixation. Animals were processed using the FISH protocol with a modified EdU labeling step after probe hybridization. Animals were incubated in the dark for 30 min in an azide click reaction containing 1% 100 mM $CuSO_4$, 0.1% 10mM azide-fluorophore 545 (Sigma #MKCH3642), 20% 50 mM ascorbic acid from (+) sodium-L-ascorbate in PBS. Ascorbic acid was made fresh for each reaction. After EdU labeling animals were washed in PBSTx 3x and FISH protocol was continued.

## Quantification and statistical analysis

Statistical analyses were performed using the Prism software package (GraphPad Inc., La Jolla, CA). Comparisons between the means of two populations were done by a Student's t test. Comparisons of means between multiple populations were done by one-way ANOVA test followed by Dunnett's multiple comparison test was used when analyzing more than two conditions. Significance was defined as p < 0.05. Statistical tests, significance, data points, error bars and animal numbers (n) for each figure are provided in the legends.

## qPCR

Total RNA was isolated using Trizol (Life Technologies) from animal fragments. *activin-2* primers were designed to include both a portion of the dsRNA construct and the rest of the endogenous gene in order to avoid amplification of dsRNA. Primers for *activin-2* amplification are as follows: tccaatcatgcttctcaaagga and tcaactggattggccataattg. The SuperScript III Reverse Transcriptase kit (Invitrogen) was used to create cDNA from 500 ng of RNA input. CT values were calculated as the average of three technical replicates. These values were then normalized to the housekeeping gene *g6pd* (S2A Fig) or *gapdh* (S5B and S5F Fig) in order to calculate the ΔCt. -ΔΔCt values were calculated by measuring the difference of a given ΔCt to that of control replicates. $2^{-\Delta\Delta Ct}$ values are plotted as data in a bar graph, along with the standard deviation for each set of values.

## Supporting information

**S1 Fig. Phylogenetic analysis of *Schmidtea mediterranea activin-2*.** Related to Fig 1. (A) Phylogenetic tree for the placement of *Schmidtea mediterranea activin-2*. Bayesian analysis of TGF- superfamily ligand proteins with a focus on Activins and Myostatins, where BMP-2/4 is used as an outgroup across species. Percent posterior probability is indicated at nodes. Smed (*Schmidtea mediterranea*), Nve (*Nematostella vectenesis*), Mmu (*Mus musculus*), Bfl (*Branchiostoma floridae*), Sko (*Saccoglossus kowalevskii*), Lgi (*Lottia gigantea*). In *Mus musculus* genes that contribute to Activin proteins are called *inhibin*, we used this nomenclature in the tree. Protein sequences are provided in S1 Data.
(PDF)

**S2 Fig. Characterization of the *activin-2* RNAi phenotype.** Related to Fig 1. (A) RNAi control. Transcript abundance for TGF-β ligands and *follistatin* from bulk sequencing expression data in control and *activin-2* RNAi animals. RPM is reads per million. Data is three replicates of a post-pharyngeal fragment. Related to S2 Data. (B) Regenerating *activin-2* RNAi animals after 40 days of RNAi, and 14 days post amputation. The three representative images show examples of (Left) anterior axis bifurcation, (Middle) parapharyngeal bulging indicative of ectopic pharyngeal tissue; (Right) posterior axis bifurcation and loss of polarity resulting in ectopic posterior heads. (C) Negative results related to 1C. FISH shows (left) CNS (*chat*[+]) and mouth and esophagus (*NB.22.1e*) and (right) PCG expression (midline *slit*[+]; anterior *sFRP-1*[+]). Yellow arrow shows ectopic posterior mouth tissue. (D) DAPI shows intact (Left) and regenerated (Right) animals develop ectopic pharynges. Red arrows show anatomy. Phx = pharynx. (E) Intact *activin-2* RNAi animals develop ectopic pharynges (Top) and ectopic mouth tissue (Bottom). At 21 days RNAi, 5/7 animals display an anteriorly expanded dd_554 domain, and a 6/7 display posteriorly expanded NB.22.1e domain. At 40 days RNAi, all animals assayed display ectopic pharynges and mouth tissue. (F) Ectopic pharynges express the transcription factor *foxA*, dd_554, and are connected to the (6G10+) muscularized gut by an NB.22.1e[+] esophagus. 6G10 is a muscle antibody, other markers are RNA probes. (G) *activin-2* expression at 0 and 24 hours post amputation in a midbody piece. (H) t-SNE representation of clustered cells (dots) colored according to single gene expression or cluster assignment based on global gene expression. (Top left) 50,562 cells [50] obtained by Drop-seq are colored according to major planarian tissue type cluster assignment. (Top right) Each cell is colored and sized by the average normalized expression of *activin-2*. (Bottom) Muscle cells obtained by Smart-seq2 [51] are colored according to planarian muscle cluster assignment. *activin-2* expression is plotted on these cells. (I) (Top left) *activin-2* is expressed in *collagen-2*[+] muscle cells in the bodywall (Bottom left) *activin-2* is expressed in *mp-1*[+] muscle cells in the pharynx (Top right)

*activin-2* is expressed in the intestine. (Bottom right) *activin-2* is highly expressed in an inner muscle wall of the pharynx, interior to *slit*⁺ cells. White shows co-localization. Scale bars represent 200 μm, except for high magnification panels in S1F (Left) where they represent 10 μm. (PDF)

**S3 Fig. Characterization of the *activin-2* RNAi intact animal pattern.** Related to Fig 2. (A) Graphs show body width/length per animal at different time points. Width quantified at widest point of the animal. (B) Intact *activin-2* RNAi animals display proper PCG expression and pole presence at 21 days of *activin-2* RNAi along the AP axis by FISH. (Pink) Anterior restricted genes, (Green) posterior restricted genes. *act-2* is abbreviation for *activin-2*. (C) Quantification of PCG domain expression relative to animal length. The length to body ratio for *ndl-3* and *wntP-2* was blind scored following the landmarks show in the top left. Data are presented as mean +/- SD. A student's t-test was performed to determine significance with a cutoff of p = 0.05. (D) Intact *activin-2* RNAi animal midline (*slit*⁺) gene expression at 21 days and 40 days by FISH. (E) Intact animal muscle fibres (6G10+) at 21 days and 40 days *activin-2* RNAi by immunofluorescence. 40 day RNAi animals display loss of orthogonal directionality of muscle fibres. (F) Intact *activin-2* animal muscle cell gene expression (*collagen-2*) at 40 days of *activin-2* RNAi by FISH. (G) Live images of intact control and *activin-2* animals at 60 days of RNAi. (H) (Left) Immunofluorescence (6G10) of control and *activin-2* animals showing muscle fibers, (Middle, Right) FISH of control and *activin-2* RNAi animals showing central nervous system *(chat)*, and PCGs (*ndl-3*, *wntP-2*) at 60 days of RNAi. (I) Intact *activin-2* RNAi animals display maintenance of restriction of the anterior pole maker *notum* at 60 days RNAi by FISH. (PDF)

**S4 Fig. Characterization of the *activin-2* RNAi animal wound response.** Related to Fig 3. (A) Heatmap of top 100 muscle specific genes annotated in [19] from bulk sequencing of posterior-facing wounds at 0 hours post amputation after 21 days of either *activin-2* or control RNAi (three wound sites per replicate). Each gene is a row, and each replicate is a column. Related to S2 Data. (B) Heatmap of wound induced gene expression from bulk sequencing of posterior-facing wounds at 0, 6, 18, 24, and 48 hours post amputation, and anterior-facing wounds at 18 hours post amputation (three to four wound sites per replicate). Each gene is a row, and each replicate is a column. Related to S2–S4 Data. Top: known wound induced genes expressed in muscle (*notum*, *foxD*, *fst*, *inhibin-1*, *wntless*, *nlg-1*, *wnt1*), epidermis (*hadrian*), neoblasts (*runt-1*), and broadly induced (*fos-1*). Bottom: known neoblast genes, expression increases indicate an increase in cycling cell number (C) Left: *in situ* hybridization of *activin-2* at 18 hpa, Right: carton showing DE-Seq result from bulk sequencing data that there is no significant difference in *activin-2* expression at 18 hpa between anterior- and posterior- facing wounds in control animals. (D-E) Wound-induced gene expression at 18 hours post amputation. Magenta denotes anterior-facing wound, and green denotes posterior-facing wound. (D) FISH shows genes that are expected to change given DEseq data. (F) Wound induced *foxD* expression is correlated to midline (*slit*⁺) width. *activin-2* RNAi animals have a marked increase in midline (*slit*⁺) width by 35 days, and *foxD* expression at 6 hours post amputation is increased to a similar width (ventral view). (G) *foxD* expression in *activin-2* RNAi animals is specific to longitudinal muscle (*myoD*⁺, *snail*⁺) at posterior-facing wounds by FISH. Pink number is number of *myoD/snail*⁺ cells co-localized with *foxD*. Yellow number is total *foxD*⁺ cells. White number is percent *foxD* cells that are longitudinal fibers. Scale bars represent 200 μm, except for the high magnification panel in S4G where they represent 10 μm. (PDF)

**S5 Fig. Characterization of symmetric wound-induced *notum* as turnover dependent.** Related to Fig 5. (A) Wound-induced *notum* expression at posterior-facing wounds first appears after 21 days of RNAi (18 hpa) by FISH. (B) RT-PCR quantification of *activin-2* expression at 7, 14, and 21 days of *activin-2* or control RNAi. Relative expression is plotted as 2-ΔΔCT values. Data are plotted as mean ± S.D. NS if p>0.05. (C) Zoom out of EdU labeling with notum FISH seen in Fig 5A. Anterior- and posterior-facing wounds of *activin-2* RNAi are shown. Animals were fed with EdU once at day 15 of *activin-2* RNAi. (D) *smedwi-1* (neoblast marker) gene expression in different irradiation conditions performed in Fig 5B and 5C. Left: control with no irradiation, Middle: 18 days post 1350 rads, Right: three days post 6000 rads. (E) *activin-2* gene expression across conditions performed in Fig 4D and 4E. (F) RT-PCR for of *activin-2* for RNAi condition stated. Relative expression is plotted as 2-ΔΔCT values. Data are plotted as mean ± S.D. NS if p>0.05. Magenta denotes anterior-facing wound, and green denotes posterior-facing wound. Scale bars represent 200 μm.
(PDF)

**S6 Fig. Ectopic anterior fates and ectopic *notum* is locally expressed.** Related to Fig 6. (A) Number of pole cells (*notum*$^+$ and/or *foxD*$^+$) of (Left) intact animals at 21 days RNAi, (Middle) intact animals at 40 days RNAi, and 72 hpa (hours post amputation) animals at 40 days of RNAi. Student's t-test used to determine statistical significance. NS is >p 0.05 *act-2 = activin-2*. (B) FISH of poles (*notum*$^+$ or *foxD*$^+$) of intact animals at 40 days RNAi. (C) Quantification for Fig 6A. Width of control and *activin-2* RNAi animal poles (40 days of RNAi) at 72 hpa. Each datapoint is width of pole/width of wound face. NS >0.05. (D) *notum* expression at wounds in a head fragment and a midbody fragment after six weeks of *nkx1.1* (circular fiber transcription factor) RNAi. Green arrow shows elevation of posterior-facing *notum* at 18 hours post amputation. (E) Immunofluorescence (6G10) showing muscle fibers at posterior-facing wounds at 48 hpa. (F) Live images of animals showing wound contraction at 30' post amputation. Holtfreter's solution inhibits muscle contraction, and was used as a negative control.
(PDF)

**S7 Fig. Ectopic anterior tissue forms at posterior-facing wounds in *activin-2* RNAi.** Related to Fig 6. (A) FISH shows expression of anterior pole marker (*notum*), and posterior pole marker (*wnt1*) in the same posterior-facing wound (Left: also shows anterior-facing wound) of an *activin-2* RNAi regenerating fragment at 30 hours post amputation. Left: Magenta arrows show *wnt1* cells proximal to posterior-facing wound. Green arrows show *notum+* cells. White arrows denote cells that co-express *notum* and *wnt1*. 19 total animals were examined with varying FISH patterns; representative images of this experiment are shown here and in 7C. (B) FISH shows expression of anterior pole marker (*notum*), anterior PCG (*sFRP-1*), and posterior PCG (*wntP-2*) in the same posterior-facing wound (B). (C) FISH shows expression of anterior pole marker (*notum*), and posterior pole marker (*wnt1*) in the same posterior-facing wound of an *activin-2* RNAi regenerating fragment at 48 hours. (D) Heatmap shows posterior PCGs and markers from bulk sequencing of posterior-facing wounds at 0, 6, 18, 24, and 48 hours post amputation, and anterior-facing wounds at 18 hours post amputation. Each gene is a row, and each replicate is a column. Related to S2 Data. (E) *foxD* and *notum* are co-expressed in the ectopic anterior pole at the posterior-facing wound of an *activin-2* RNAi animal fragment at 72 hours post amputation. (F) Lower magnification of image shown in Fig 7C denotes close proximity between *ovo*$^+$ eye progenitors and the anterior PCG *sFRP-1* at a posterior-facing wound of an *activin-2* RNAi animal at 48 hours post amputation. Scale bars, 200 μm.
(PDF)

**S1 Data. This file contains a list of protein sequences used in the phylogenetic analysis presented in S1 Fig.** Each sequence is annotated with species and available information of deposited location. This file also contains the trimmed input for Bayesian analysis from row 21–68.
(XLSX)

**S2 Data. This file contains bulk RNA-sequencing analysis for regenerating control and *activin-2* RNAi animals.** The DESeq tool was used to calculate log2FC and $p_{adj}$ for each condition. Each transcript is annotated with best BLAST hits.
(XLSX)

**S3 Data. This file contains a list of contigs that are the union of FISH validated wound induced genes found in Wurtzel et al 2015 and Wenemoser et al 2012.** Each transcript is annotated with best BLAST hits.
(XLSX)

**S4 Data. This file contains a list of contigs that are the genes from S3 Data which significantly differ in S2 Data specifically at times 6 hpa and/or 18 hpa.** The cutoff for this table is $p_{adj} < 0.001$. Rows from S2 Data meeting these specifications were transposed into this table. Each transcript is annotated with best BLAST hits.
(XLSX)

## Acknowledgments

We thank members of the Reddien lab for discussions and comments on the manuscript.

## Author Contributions

**Conceptualization:** Jennifer K. Cloutier, Peter W. Reddien.

**Data curation:** Jennifer K. Cloutier, Conor L. McMann, Isaac M. Oderberg.

**Formal analysis:** Jennifer K. Cloutier, Conor L. McMann, Peter W. Reddien.

**Funding acquisition:** Peter W. Reddien.

**Investigation:** Jennifer K. Cloutier, Conor L. McMann, Isaac M. Oderberg.

**Methodology:** Jennifer K. Cloutier, Conor L. McMann, Peter W. Reddien.

**Supervision:** Peter W. Reddien.

**Validation:** Jennifer K. Cloutier, Conor L. McMann.

**Visualization:** Jennifer K. Cloutier, Conor L. McMann.

**Writing – original draft:** Jennifer K. Cloutier, Peter W. Reddien.

**Writing – review & editing:** Jennifer K. Cloutier, Conor L. McMann, Peter W. Reddien.

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
