## [Decision Letter · Decision Letter 0]

1 Jun 2020

Dear Dr Reddien,

Thank you very much for submitting your Research Article entitled 'activin-2 is required for regeneration of polarity on the planarian anterior-posterior axis' to PLOS Genetics. Your manuscript was fully evaluated at the editorial level and by independent peer reviewers. The reviewers appreciated the attention to an important problem, but raised some substantial concerns about the current manuscript. Based on the reviews, we will not be able to accept this version of the manuscript, but we would be willing to review again a much-revised version. We cannot, of course, promise publication at that time.

If you decide to revise the manuscript for further consideration at PLOS Genetics, please aim to resubmit within the next 60 days, unless it will take extra time to address the concerns of the reviewers, in which case we would appreciate an expected resubmission date by email to plosgenetics@plos.org.

[LINK]

We are sorry that we cannot be more positive about your manuscript at this stage. Please do not hesitate to contact us if you have any concerns or questions.

Yours sincerely,

A. Aziz Aboobaker

Associate Editor

PLOS Genetics

Gregory P. Copenhaver

Editor-in-Chief

PLOS Genetics

Reviewer's Responses to Questions

**Comments to the Authors:**

Reviewer #1: Cloutier, Oderberg, and Reddien

Activin-2 is required for regeneration of polarity on the planarian anterior-posterior axis

PGENETICS-D-20-00448

In this manuscript, Cloutier and colleagues investigate the mechanism through which planarian Activin-2 affects body polarity and regeneration. Building upon prior work from the Reddien group and others, this manuscript shows that activin-2 RNAi causes pleiotropic phenotypes that sometimes include ectopic heads in the posterior of the animal. The authors further show evidence that notum expression is not asymmetrical at 18 hours post-amputation in activin-2 RNAi animals, which may lead to subsequent polarity defects. This work is thorough, with well-designed experiments and beautiful images and thorough quantification throughout. The work adds to our understanding of planarian polarity signaling, particularly after injury. The authors present a strong body of work that should be interesting to the readers of PLoS Genetics. The manuscript should be acceptable for publication once the following concerns are addressed. Nearly all of the listed concerns should be addressable with writing changes or potentially quantification or inclusion of existing results.

Major concerns:

1. At times, the authors misrepresent the novelty of their contribution in this manuscript. For example, they make claims like “Activin signaling was previously not known to regulate regeneration polarity” and “Here we report an unexpected role for Activin signaling in controlling head-versus-tail regeneration in planarians.” They also declare that they are showing a “novel” activin-2 RNAi phenotype. These claims are somewhat misleading. Activin-2 has been previously characterized in Gavino, et al (eLife 2013) and Roberts-Galbraith and Newmark (PNAS 2013) (activin-2=activin in the PNAS paper). Undoubtedly, this new manuscript reports additional phenotypes (ectopic posterior heads), probably due to better penetrance of the RNAi or differences in experimental timing. Additionally, this manuscript reports a more detailed mechanism for this phenotype and takes the prior findings in exciting new directions. But throughout the manuscript, the authors should be clearer about what is new in this work and how it relates to prior research. In particular, the data presented in the PNAS paper implicated Activin signaling in polarity and in inhibition of anterior fates, so this manuscript ties nicely into the prior model.

2. The data that are used to support the argument that Activin-2 is important in Notum asymmetry are sometimes a bit unclear.

A) First, the data presented only support symmetrical notum expression after activin-2 RNAi at 18h post-amputation (Supp. Fig. 4A). Is that correct? I think that the average reader might reasonably conclude that Notum expression is symmetrical throughout regeneration after activin 2 RNAi (e.g. “We conclude that Activin has an essential role in the asymmetric activation of notum… during planarian whole-body regeneration.”). The authors should consider moving the data from Supp. 4A into the main text to avoid this confusion, perhaps with more quantification of the symmetry of expression over time. Words like “transient” to remind the reader of the temporal nature of the phenotype might also be helpful.

B) The data in 3B are also a bit confusing, perhaps because Notum is expressed predominantly in the anterior (normally) and thus posterior RPM is hard to interpret in context. Are data available to show both posterior and anterior RPM (and/or potentially the anterior:posterior ratio) over time?

C) It seems that there is a bit of a logical disconnect between the transience of the symmetrical Notum phenotype and the striking nature of the long-term phenotype (including reexpression of posterior Notum in posterior heads, Fig. 6B). If Notum expression (or rather the symmetry of Notum expression) is back to normal by 24 h post-amputation, how do the authors think that the longer-term phenotypes arise? I think this disconnect could probably be addressed in the text or even in the discussion to help a reader connect the different elements of the phenotype in one coherent model. The model figure (Fig. 7) seems to omit a complexity of the story if polarity is symmetrical at 24 hpa.

3. The authors argue that the activin-2 phenotype with regards to polarity is “regeneration-specific.” However, the authors also state that “subtle anterior shifting of wntP-2 expression domain length could not be excluded.” Have the authors attempted to quantify the expression domains for ndl-3, wntP-2, and ndl-5 (e.g. head to anterior margin of the domain, length of the domain versus length of the animal) at day 20 post-amputation? The images do look somewhat different, which might mean there is also a homeostatic phenotype. Have the authors looked at AP gradient markers at later time points (e.g. 6 weeks). By that time point, there is a dramatic change in animal shape as well as a change in slit expression (Supp. Fig. 2) which indicates that ML polarity is affected homeostatically. My expectation is that AP might be affected then more strongly than at 3 weeks/20 days. If AP effects are not exclusive to regeneration, the language around this point probably need to be altered.

Minor concerns:

1. The authors show that activin-2 RNAi causes muscle disorganization and slit misexpression. The Activin pathway (at least the receptor) has also been shown to have a role in fissioning behavior (Arnold, 2019). Did the authors note any changes in behavior/movement in the activin 2 RNAi animals? Is it possible that muscle function is perturbed in these animals and – if so – could wound closure be affected? If failed would closure results in a wider or more uneven starting point for regeneration, could this contribute to splitting of heads/tails?

2. Did the authors try to “rescue” split heads or tails with slit(RNAi) to determine if the slit domain expansion was causative in other phenotypes?

3. In Fig. 1B, there seems to be extra chat staining in the anterior of the animal (in the middle of the head). Is this often seen in these animals? Do the authors think that this is out of place pharyngeal tissue, brain tissue, or something else? This might be another part of the improved phenotype worth mentioning.

4. The authors do not show expression of activin-2 after pre/postpharyngeal amputation and 18hpa. Is there evidence that activin-2 expression is asymmetric at 18 h? I think this information would be helpful in imagining how activin-2 might affect notum, but I would not recommend holding up the paper for this experiment, given current lab shutdowns for COVID-19.

5. Can the authors please clarify in the figure legend which animals were used for quantification in Fig. 1C? The denominators are not the same, so I think some stained animals were used for some but not all quantification, but I can’t be sure.

6. Is pigmentation affected by long-term activin-2 RNAi (Fig. 2A)?

7. Can the authors clarify the dosage of dsRNA in the methods (concentration or total mass)? Given that this work shows new phenotypes, potentially due to RNAi effectiveness, dosage information would be helpful.

8. The data from irradiation experiments were a bit challenging to interpret, especially since the time points post-irradiation are not 18 hpa. The result in Fig. 5B indicates that the asymmetry phenotype is stem cell independent. But the sub-lethal irradiation experiment with a time frame in which stem cells are largely recovered (Supp. 5C) shows no symmetric expression. Is the argument that sublethal irradiation prevents activin 2 RNAi animals from misspecifying muscle (or accumulating disorganized muscle) and then without muscle disorganization you don’t see symmetric notum expression? I think that, particularly for non-expert readers, the take-home message for these experiments could be clarified.

9. For RNAi experiments in which an interesting phenotype is seen in a minority of animals (e.g. Fig. 1B, 6C, 6E), it would be helpful to include the phenotypes that are most prominent, as well. This will help the reader to interpret data properly and get a feel for the full range of RNAi phenotypes.

Reviewer #2: In the manuscript by Cloutier et al., the authors investigate the roles of activin-2 in establishing regeneration polarity upon regeneration in the planarian S. mediterranea. Following transverse amputation of a planarian, a regeneration polarity decision must be made to appropriately regenerate a head at anterior-facing wounds and a tail at posterior-facing wounds. The expression of notum, a wnt signaling inhibitor, is the first indication of a differentiation between the anterior and posterior wound sites, as notum is preferentially expressed at anterior-facing wounds. This work demonstrates that following knockdown of activin-2, amputated worms regenerate ectopic heads at posterior facing wounds. Activin-2 knockdown worms also experience axis bifurcations at anterior and posterior regenerating blastemas. RNA sequencing analysis shows that activin-2 knockdown worms have symmetric notum expression at anterior and posterior wounds, but no other functionally significant alterations in early wound response gene expression. Interestingly, the authors show that production of new longitudinal muscle cells is required for this symmetric notum expression, suggesting that activin-2 exerts its effects during muscle cell differentiation. Later in regeneration, activin-2 RNAi worms express both anterior and posterior positional control genes at posterior-facing wounds, resulting in the generation of discrete anterior structures in the posterior. This suggests that activin-2 restricts wound-induced notum expression to anterior-facing wounds to promote tail regeneration at posterior-facing wounds and that activin-2 is a regulator of regeneration polarity. This work identifies the first regulator of asymmetric notum activation, providing important insight into the question of how planarians differentiate between anterior and posterior wounds. The data are high quality, however, some key experiments are missing, and/or over-interpreted. While the phenotypes are of interest, the mechanism for the most interesting phenotype, the axis bifurcation, is not thoroughly investigated.

Major Concerns

1. In the Introduction, a summary of known follistatin and activin and TGFB phenotypes known in planarians so far is warranted (which is significant). The current lines about activin and follistatin in planarians (lines 103-107) are vague and not helpful to the reader put your study into context of what is known and what is missing.

2. Need proper phylogenetic analyses of the TGFB family to resolve whether planarian activins are activins or myostatins in order to resolve exactly the issues raised in lines 122-128.

3. Line 168: seems like an over-interpretation to fit the authors “story” as opposed to objectively stating the reality that activin-2 is detected in every major cluster in scRNAseq, and in muscle subclustering, high expression was seen in DV-like and a sub-cluster of circular (not in all circular muscles as described and annotated in S1F).

4. The role of symmetric notum expression in the regeneration of ectopic posterior heads is compelling. However, the association between symmetric notum activation and axis bifurcation during regeneration is still unclear. The model figure as well as the nkx1.1 experiments seem to suggest that axis bifurcation in both anterior and posterior blastemas occurs as a result of symmetric notum activation, but as notum activation in the anterior is normal in activin-2 knockdown worms, it is unclear how bifurcated blastemas form in the anterior. Do the sublethally irradiated worms from Fig 5C eventually regenerate? If so, do the activin-2 knockdown worms with asymmetric notum activation still develop bifurcated blastemas?

5. There appears to be a larger number of cells expressing notum in uninjured activin-2 knockdown worms (Fig 2B) which was not discussed. This raises the question of whether there is a difference in the number of muscle cells expressing positional control genes during regeneration, and whether this contributes to the apparent increase in notum expression contributes to the bifurcations upon regeneration. In the cases where the anterior blastema regenerates as normal, is the number of pole cells normal as well? Quantification of foxD+ pole cells would address this issue across all phenotypes.

6. This paper mentions that Follistatin is required for the missing tissue response and is a regulator of Activin, but did not explore the role of Follistatin in regulating activin-2 during regeneration. The authors report that the missing tissue response is normal in activin-2 knockdown worms based on the expression of neoblast genes in RNAseq (lines 222-227). This should be supported with quantification of proliferation during the first 2dpa. Additionally, if Follistatin does act to inhibit activin-2 as well as activin-1, double RNAi of follistatin with nkx1.1 could be used to test the hypothesis that the bifurcations seen in nkx1.1 knockdown worms are due to a decrease in activin-2 expression.

7. A more in-depth analysis of the muscle cell subsets involved in the activin-2/notum response to injury would be beneficial, particularly given the extensive work previously done by this group on muscle cell subsets. The claim that newly-formed longitudinal muscle fibers are responsible for asymmetric notum activation could be better supported using myoD knockdown worms, where differentiation of new longitudinal muscle fibers is blocked (assuming it is feasible to generate myoD/activin-2 RNAi worms).

Minor Concerns

1. Structural issues (citing fig S3A before S1F). Never mentioning the top of S1F.

2. S1F would be helpful to be next to Fig 1D (or move 1D to supplemental).

3. The implications of the irradiation experiments were not fully discussed in the text. Do these results suggest that longitudinal muscle fibers have anterior/posterior orientation ‘encoded’ by activin signaling during differentiation? This is an interesting point that warrants further discussion.

4. 40 days of RNAi treatment was used to determine the effect of activin-2 knockdown at homeostasis. This ruled out other phenotypes excluding multiple pharynges in uninjured activin-2 knockdown worms. This time frame does not seem long enough to allow for sufficient tissue turnover.

5. Figure 3C: the RNA sequencing experiments are conducted at different time points with tissue at different amputation sites. It was unclear why the different tissue fragments were analyzed, as this was not addressed in the text.

6. The authors state that some genes changing in RNAseq do not look different by FISH in Fig. 3D. However, the images shown look substantially lower in the posteriors of activin-2 RNAi for fst, inhibitin, wnt-1, and wntless, while nlg-1 looks substantially higher.

7. The control and activin-2 RNAi images in Figure 5B are placed in opposite order to the rest of the figures (i.e. control on the right instead of on the left), which is confusing.

Reviewer #3: In the manuscript entitled ‘activin-2 is required for regeneration of polarity on the planarian anterior-posterior axis’ Cloutier et al. report a very inspiring phenotype obtained after silencing Activin-2 in planarians. The authors show that activin-2 RNAi resulted in the regeneration of ectopic posterior heads following amputation. Importantly, they observe that notum, the main element of the Anterior signaling center, is not downregulated at 18h in P wounds, providing a molecular explanation for the ectopic posterior-heads. Activin-2 RNAi animals also showed AP axis splitting, and this was specific of regenerating animals, as it did not occurred during normal homeostasis. The authors conclude that Activin-2 could be one of the signals coming form the pre-existing tissue that controls notum expression (Wnt signalling levels) and thus to date it would be the earliest known step in establishing head-versus-tail identity.

The study is of general interest, as it boards general and important questions related with regeneration and tissue patterning. The experiments are properly planed and in general well exposed and justified. However, some interpretations could be misleading, and a deeper analysis and discussion of the main finding, that is, the maintenance of notum expression in P and the regeneration of poles with different identities, should be performed.

- The authors show that in Act-2 RNAi animals notum is not downregulated at 18h in P, and propose that it could be the cause of the multiple heads/tails in P. They also have some evidences that it could be related with the integrity of circular and longitudinal fibers, according to previous published results (Scimone et al. 2017). However, the present study lacks a more in deep analysis and discussion of the mechanism underlying this phenotype. How can it be that from a homogeneous expression of notum in the 18h P wound, few hours later different A and P organizing centers appear? First, it needs a more detailed description and quantification of the phenotypes, and second, some more experiments could be performed in order to explain how the increase of notum in P at 18h leads to multiple organizing centers with different identity. For instances, the timing of expression of not only notum but wnt1 during P regeneration (only 48 h are shown) could give some clues, as well as the analysis of the longitudinal and circular fibers at the region that must regenerate.

-In the abstract it is stated that ‘Activin-2 is required for this head-versus-tail regeneration decision’. And this appears to be a main conclusion of the study. However, the phenotype shows that Activin-2 seems to be required to restrict a unique axis, but not to decide the identity of the poles. The results show that notum is not downregulated in P, but in fact a tail is regenerated. The interpretation of the results should be more linked to the real observations.

-In the first section and the corresponding Figure 1 the authors describe the appearance of a ‘variable numbers of heads and tails in fragments with both head and tail amputated’. And they show a quantification in Figure 1C. The description and the quantification of the phenotypes observed must be more specific to really understand what is happening in Act-2 RNAi animals. What are really the buds they have in P, or in lateral positions? (SF1). They need to use markers of P and A identity. And how many tails, heads, o tail and heads, appear in P? 2, 3? When there are 3, the one in the middle is always P and the 2 lateral are A? When there are 2, each one has different identity? It’s necessary to show a detailed description and quantification to clarify this point, since it’s important to understand to which extent Act-2 has a role in polarity, or has a role in controlling notum, or in restricting the P organizing center… In fact, these are possibilities that are not properly discussed in the manuscript.

-In the quantification in Figure 1C it seems that A axis bifurcation takes place much later than P axis bifurcation. Why is it like that? In fact, in Figure 6A it is shown that at 72h notum expression is already splited in 2. Thus, why in the graph in Fig 1C there are so few bifurcated heads at 14-21 days? And why the number of splitted A heads increase with time? It should be discussed.

-Supp Fig 1B- In this experiment the animals have been regenerating for 14dpa, but they have been inhibited for 40 days, so the effect seen in the pharynges are due to tissue renewal, not to tissue regeneration. May be also the lateral buds. When analyzing regeneration, the timing of RNAi and amputation must be taken into account, otherwise one could take wrong conclusions. Furthermore, what is the identity of the lateral buds? Is this a common feature of the phenotype?

-The finding that during homeostasis polarity is not affected is very relevant, and it is not properly discussed.

-The analysis of the RNAseq is confusing. 14 genes displayed significantly different expression at 6 -18 hours but the authors argue that the analysis by FISH shows no differences. Where is it this FISH analysis? The genes in Figure 3D do not correspond to the ones in Table 2. And furthermore, if RNAseq analysis shows a differential expression, this result is more quantitative than a FISH, isn’t’ it?

-In Figure 3D, the authors conclude that there is no difference in the expression of those genes, but apparently wnt1 and wntless seem to be downreglated in P wounds in Act-2 RNAi animals. This result would be important for the study, since upregulation of notum could came together with downregulation of wnt1. This is a very important point that should be clearly solved.

-The conclusion of the RNAseq section is that ‘Of all wound-induced genes assessed by FISH and RNA sequencing, only notum was affected at the time point when AP regeneration polarity defects emerged following activin-2 RNAi’. This is a strong conclusion that, according to the previously exposed, lacks more supportive data.

-The RNAseq results show a very interesting result: notum appears to be expressed in P, and thus, coexpressed with wnt1, but it needs to be downregulated at 18h to make a tail. This result must be discussed.

- The authors show that the loss of notum polarity in activin-2 RNAi animals was observed at 21 but not at 7 and 14 days post-RNAi initiation. They hypothesize that ‘activin-2 could be required during muscle cell turnover to maintain regeneration polarity.’ What does it exactly mean? That Act-2 could be necessary for maintenance of the longitudinal/circular fibers integrity? In fact, the defects observed during homeostasis could fit with this hypothesis. But then, to test if this is true the authors should 1) see if after 7 -14-21 days of RNAi, the mRNA levels of act-2 are really downregulated at the same levels, and if it’s the case, then 2) analyze whether the longitudinal/circular muscles are differentially affected in the 3 situations in the region that will be amputated.

The irradiation experiments do not seem to clarify much the mechanism. In Figure 5B- is really the first image Act-2 RNAi, or it is the control, as in the rest of images? In any case, the conclusion is that notum is still expressed after 6000 rads in P, so at 16h the expression of notum does not depend on neoblast. This could be expected. At this timepoint some notum expression could be neoblast dependent and some could be neoblast independent. But what happens at 18h and later? This should be analyzed. Then after several days of low irradiation notum in P is not expressed anymore in act-2 RNAi animals, but what is the conclusion? That a healthy muscle is necessary to regenerate? How is the muscle in these animals? How are the other organs? May be the digestive system is the one related with notum expression, since irradiation affects all tissues.

Additional comments:

Lines 125-126. The authors refer to Kenny et al. for the classification of Activin-2. However, in this study Smed Activins are not included. A specific phylogegentic study of Smed Activins/myostatins should be cited or performed. It is important to be clear about the identity of the activin-2 that is the focus of the study. Even more, considering that in the introduction a comparison with activin/follistatin function in other systems is exposed, to suggest its function in animal regeneration.

Lines 134-137- The authors assume that multiple tails or heads appear at P, but in fact at this point of the study any P marker is analyzed, so the identity of the regenerated fragment cannot be really assessed.

Line 166- Figure S3E should be corrected to Figure S1E

Line 192- It reads after 60 days of RNAi but in the figure legend it reads at 40 days. What is the correct?

The interpretation of the graph in Figure 3C is really hard.

A scheme showing the RNAi and amputation timing of each experiment would be helpful.

Do changes in cell death or proliferation could give some clues about the function of Activin-2 in axial restriction?

**Have all data underlying the figures and results presented in the manuscript been provided?**

Reviewer #1: Yes

Reviewer #2: No: RNAseq data will be released after publication.

Reviewer #3: Yes

PLOS authors have the option to publish the peer review history of their article (what does this mean?). If published, this will include your full peer review and any attached files.

Reviewer #1: No

Reviewer #2: No

Reviewer #3: No

---

## [Decision Letter · Decision Letter 1]

21 Jan 2021

Dear Dr Reddien,

Thank you very much for submitting your Research Article entitled 'activin-2 is required for regeneration of polarity on the planarian anterior-posterior axis' to PLOS Genetics.

The manuscript was fully evaluated at the editorial level and by independent peer reviewers. The reviewers appreciated the attention to an important topic but identified some concerns that we ask you address in a revised manuscript

We therefore ask you to modify the manuscript according to the review recommendations. Your revisions should address the specific points made by each reviewer.

[LINK]

Yours sincerely,

A. Aziz Aboobaker

Associate Editor

PLOS Genetics

Gregory P. Copenhaver

Editor-in-Chief

PLOS Genetics

Reviewer's Responses to Questions

**Comments to the Authors:**

Reviewer #1: Cloutier, McMann, Oderberg, and Reddien

Activin-2 is required for regeneration of polarity on the planarian anterior-posterior axis

PGENETICS-D-20-00448

Most concerns from the prior review were adequately addressed.

Minor concerns:

1. I think that the authors have made some effort to put the work into prior context. I do recommend that the authors clearly state that their activin-2 is the same as activin in a prior publication, so that readers could better evaluate how this paper fits with prior results and models. It should not be incumbent on the reader to look up sequences and supplementary information to sort this out.

2. The authors did add a concentration of dsRNA but not a total volume. Can the authors please add whether 5-8 ug/ul is the concentration of the final liver+dsRNA mix or the initial concentration of dsRNA. If the latter, then the reader may also need to know what volume of dsRNA and what volume of liver mix were used. With the methods as written, an interested reader would not be able to replicate this experiment.

Reviewer #2: In the manuscript by Cloutier et al., the authors have made substantial improvements to the manuscript and work. Particularly appreciated was the amount of experiments performed in request of all of the reviewers, especially considering the difficult times. The clarity of the manuscript, as well as the completeness of the study is much more cohesive and will be of broad interest to the Plos Genetics readership as well as the planarian field. I believe that all of my previous concerns have been addressed by the authors.

Reviewer #3: In the revised version of the manuscript Cloutier et al. have properly answered and addressed the main concerns. The manuscript has improved substantially by including more detailed analysis of the phenotypes, which were necessary for their understanding. The study is now complete, although the interpretation of the data remains confusing:

1- The result sections include several paragraphs that could be integrated into the discussion. This would make it easy for the reader to follow. Also to simplify it, the last section of the results (Local PCG expression changes in activin-2 RNAi animals were associated with ectopic anterior neoblast fate specification) could be moved to supplementary and integrated with the previous one.

2-The results indicate that Activin-2 is restricting notum to the anterior wound at 18h, which is a key stage to specify the identity of the pole in the wounds. The co-expression of wnt1 and notum in posterior wounds leads to the range of defects in posterior regeneration described in the study. The problem is the conclusion about the mechanism of action of Activin-2 . The authors conclude that Activin-2 could be necessary for the establishment of a ‘polarity’ in the longitudinal fibers, which are the ones expressing notum. The main reasons underlying this conclusion are 1) that according to their interpretation, the muscular fibers do not appear affected in the activin-2 RNAi animals; and 2) inhibition of myoD+Activin2 avoids the ectopic expression of notum in posterior. However, 1) in Supp Fig 6 E the muscular fibers do not seem properly organized at 48h of regeneration in Activin-2 RNAi animals (in 21 days RNAi animals they could already be affected according to supp figure 3D). Regarding point 2) myoD inhibition decreases notum expression in anterior and posterior wounds. The quantitative difference between them could be a matter of timing, since, as the authors explain, the expression of notum in posterior wounds is delayed with respect to anterior, and this is a crucial point in the current interpretation of the data. According to the data, an alternative explanation could be that activin-2 is required for the proper regeneration of the muscular fibers (longitudinal and/or circular) and not for the establishment of a supposed polarity. This would also explain the duplication of poles seen in anterior wounds.

**Have all data underlying the figures and results presented in the manuscript been provided?**

Reviewer #1: Yes

Reviewer #2: Yes

Reviewer #3: Yes

PLOS authors have the option to publish the peer review history of their article (what does this mean?). If published, this will include your full peer review and any attached files.

Reviewer #1: No

Reviewer #2: No

Reviewer #3: No

---

## [Editor Report · Decision Letter 2]

3 Mar 2021

Dear Dr Reddien,

We are pleased to inform you that your manuscript entitled "activin-2 is required for regeneration of polarity on the planarian anterior-posterior axis" has been editorially accepted for publication in PLOS Genetics. Congratulations!

Yours sincerely,

A. Aziz Aboobaker

Associate Editor

PLOS Genetics

Gregory P. Copenhaver

Editor-in-Chief

PLOS Genetics

Comments from the reviewers (if applicable):

**Data Deposition**

http://datadryad.org/submit?journalID=pgenetics&manu=PGENETICS-D-20-00448R2

**Press Queries**

---

## [Editor Report · Acceptance letter]

21 Mar 2021

PGENETICS-D-20-00448R2 

activin-2 is required for regeneration of polarity on the planarian anterior-posterior axis 

Dear Dr Reddien, 

We are pleased to inform you that your manuscript entitled "activin-2 is required for regeneration of polarity on the planarian anterior-posterior axis" has been formally accepted for publication in PLOS Genetics! Your manuscript is now with our production department and you will be notified of the publication date in due course.

With kind regards,

Alice Ellingham

PLOS Genetics

On behalf of:
